# Sentinel-3 Delay-Doppler Altimetry over Antarctica

Malcolm McMillan[1], Alan Muir[2], Andrew Shepherd[3], Roger Escolà[4], Mònica Roca[4], Jérémie Aublanc[5], Pierre Thibaut[5], Marco Restano[6], Américo Ambrozio[7], Jérôme Benveniste[8]

[1]Centre for Polar Observation & Modelling, Centre for Excellence in Environmental Data Science, Lancaster University, Lancaster, LA1 4YW, UK
[2]University College London, Gower Street, London, WC1E 6BT, UK
[3]Centre for Polar Observation & Modelling, University of Leeds, Leeds, LS2 9JT, UK
[4]IsardSAT Ltd, Surrey Space Incubator, 40 Occam Road, The Surrey Research Park, Guildford, Surrey GU2 7YG, UK
[5]CLS, 11 Rue Hermes, Parc Technologique du Canal, 31520 Ramonville Saint-Agne, France
[6]SERCO, c/o ESA ESRIN, Largo Galileo Galilei, 1, 00044 Frascati RM, Italy
[7]DEIMOS, c/o ESA ESRIN, Largo Galileo Galilei, 1, 00044 Frascati RM, Italy
[8]ESA ESRIN, Largo Galileo Galilei, 1, 00044 Frascati RM, Italy

*Correspondence to*: Malcolm McMillan (m.mcmillan@lancaster.ac.uk)

**Abstract.** The launch of Sentinel-3A in February 2016 represented the beginning of a new long-term series of operational satellite radar altimeters, which will provide Delay-Doppler altimetry measurements over ice sheets for decades to come. Given the potential benefits that these satellites can offer to a range of glaciological applications, it is important to establish their capacity to monitor ice sheet elevation and elevation change. Here, we present the first analysis of Sentinel-3 Delay-Doppler altimetry over the Antarctic Ice Sheet, and assess the accuracy and precision of retrievals of ice sheet elevation across a range of topographic regimes. Over the low slope regions of the ice sheet interior, we find that the instrument achieves both an accuracy and a precision of the order of 10 cm, with ~98% of the data validated being within 50 cm of co-located airborne measurements. Across the steeper and more complex topography of the ice sheet margin, the accuracy decreases, although analysis at two coastal sites with densely surveyed airborne campaigns shows that ~60-85% of validated data are still within 1 meter of co-located airborne elevation measurements. We then explore the utility of the Sentinel-3A Delay-Doppler altimeter for mapping ice sheet elevation change. We show that with only two years of available data, it is possible to resolve known signals of ice dynamic imbalance, and to detect evidence of subglacial lake drainage activity. Our analysis demonstrates a new, long-term source of measurements of ice sheet elevation and elevation change, and the early potential of this operational system for monitoring ice sheet imbalance for decades to come.

## 1. Introduction

Accurate knowledge of ice sheet topography and regional changes in ice volume are essential for developing a process-based understanding of ice sheet evolution, and for monitoring the response of Greenland and Antarctica to climate change (*Davis et al., 2005; Price et al., 2011; Shepherd et al., 2004*). For the past quarter of a century, satellite radar altimeters have provided near-continuous coverage of Earth's Polar regions, yielding detailed topographic information of ice sheets (*Bamber et al., 2009; Bamber & Bindschadler, 1997; Helm et al., 2014; Remy et al., 1989; Slater et al., 2018*), together with estimates of changes in ice sheet volume (*Davis & Ferguson, 2004; Helm et al., 2014; Johannessen et al., 2005*) and mass (*McMillan et al., 2014, 2016; Shepherd et al., 2012; Wingham et al., 2006; Zwally et al., 2011*). By resolving changes at the scale of individual glacier basins, these satellites have been able to identify emerging signals of imbalance (*Flament & Rémy, 2012; Wingham et al., 2009*), loci of rapid ice loss (*Hurkmans et al., 2014; McMillan et al., 2016; Sørensen et al., 2015; Zwally et al., 2005*), and the regional contribution of ice sheets to global sea level rise (*Shepherd et al., 2012; The IMBIE Team, 2018; Zwally et al., 2011*).

During the earlier part of this 25-year record, missions carried conventional low resolution, or *pulse-limited*, instruments, including those flown onboard the ERS-1, ERS-2, Envisat and SARAL satellites. These systems, which were originally developed to measure the ocean geoid, flew to a latitude of $\sim 81°$, and provided a ground footprint of approximately 2 km$^2$ (corresponding to a Ku-band pulse-limited footprint over a flat, orthogonal surface, a 320 MHz measured bandwidth, and a satellite altitude of 700-800 km). The size of this footprint, together with the large area illuminated by the radar antenna beam ($\sim$200 km$^2$), meant that correctly locating the origin of the surface reflection in regions of complex terrain could be challenging. In 2010, the first dedicated ice radar altimetry mission, CryoSat-2, was launched, with two improvements in system design that were specifically aimed at enhancing altimeter performance in areas of steep and complex ice margin terrain. Synthetic Aperture Radar (SAR), or *Delay-Doppler*, processing delivered a four-fold improvement in along-track resolution, to approximately 400 m, and interferometric techniques were used to locate the origin of the surface reflection in the across-track plane (*Raney, 1998; Wingham et al., 2006*). These developments, in conjunction with the unique long-period, high-inclination orbit, have delivered improved coverage of Earth's ice sheets (*McMillan et al., 2017; McMillan et al., 2014*), and yielded greater confidence in determining their ongoing evolution.

With CryoSat-2 now operating far beyond its original design lifetime of 3.5 years, and the recent initiation of the Copernicus operational programme of satellites, there is presently a need to establish the utility of the new class of Sentinel-3 Delay-Doppler radar altimeters (*Donlon et al., 2014*) for the purpose of monitoring ice sheet change. The first of four satellites, Sentinel-3A, was launched on 16th February 2016, and was followed by Sentinel-3B on 25th April 2018. Each satellite provides coverage up to a latitude of 81.35°, with 385 orbital revolutions per cycle, yielding an on-the-ground revisit time of 27 days (Table 1), together with a 4-day sub-cycle. The main altimetry payload, *SRAL*, is a Ku-band SAR altimeter, which provides

elevation measurements with a resolution of ~ 300 m along-track by ~ 1.6 - 3 km across-track, depending upon the surface roughness (*Chelton et al., 1989*). To date, the focus of Sentinel-3A exploitation has been upon retrievals over ocean and inland water surfaces, with early studies demonstrating its capability to retrieve fine-scale (~20 km) oceanographic features (*Heslop et al., 2017*), to increase the quality of river level and discharge estimates in Central Africa (*Bogning et al., 2018*) and to resolve

near-coastal sea surface heights (*Bonnefond et al., 2018*).

Over ice sheet surfaces, Sentinel-3 is unique among altimeters, because it operates in Delay-Doppler mode across all regions. This mode of operation contrasts with CryoSat-2, which operates in Low Resolution Mode over the interior of each ice sheet and SAR interferometric mode at coastal locations. As a consequence, although no interferometric information is available to

aid Sentinel-3 retrievals around the ice sheet margins, high resolution measurements are for the first time routinely acquired throughout the ice sheet interior. Given these different operating modes across both inland and coastal ice sheet regions, together with the future longevity of the EU Copernicus programme of operational satellites, it is imperative that early assessments of the accuracy and precision of the instrument are made over ice sheet surfaces, to establish the basis for glaciological applications of these data. Here we provide a first evaluation of Sentinel-3 Delay-Doppler measurements over

Antarctica, to determine its utility for monitoring ice sheet surfaces.

## 2. Study Sites

To evaluate the performance of the Sentinel-3 Delay-Doppler altimeter across a range of topographic regimes, we selected four study sites across Antarctica for detailed analysis (Figure 1). Two sites – the areas surrounding Lake Vostok and Dome C (Figures 1b and 1c) – are located within the interior of the East Antarctic Ice Sheet and are characterized by relatively simple

topography. These sites allowed us to evaluate the performance of Delay-Doppler altimetry in regions representative of a large part of the Antarctic interior. Both sites have low and relatively uniform topographic slopes, with an average and standard deviation of 0.09$^\circ$ and 0.05$^\circ$ (Lake Vostok), and 0.04$^\circ$ and 0.03$^\circ$ (Dome C), respectively, based upon a 1km Digital Elevation Model (DEM) (*Slater et al., 2018*). Furthermore, the flat ice surface directly above Lake Vostok represents an established validation site for new altimetry missions (*Richter et al., 2014; Schröder et al., 2017; Shuman et al., 2006*), although it is

important to note that our study site does extend beyond the region floating in hydrostatic equilibrium, so as to incorporate areas of grounded ice with topography that is more representative of the ice sheet interior. To assess performance in regions of steeper and more complex topography, we then selected two coastal sites covering parts of Dronning Maud Land and Wilkes Land (Figure 1d and 1e). These locations were chosen because of the availability of airborne campaigns that could be used as independent validation. Both sites have an order of magnitude steeper and less uniform topography than the inland sites, with

the mean and standard deviation of the surface slope being 0.50$^\circ$ and 0.94$^\circ$ (Dronning Maud Land) and 0.40$^\circ$ and 0.51$^\circ$ (Wilkes Land), respectively. Finally, beyond these focused, site-specific studies, we also conducted several continent-wide analyses, in order to better understand the performance of the Sentinel-3A altimeter across a broader range of topographic regimes.

## 3. Sentinel-3 Data & Processing Methods

To evaluate the accuracy of Sentinel-3A elevation measurements, we begun by analysing 14 cycles of Sentinel-3A SRAL data acquired between December 2016 and December 2017. Our processing followed a standard chain, beginning with the 20 Hz waveform data provided by the European Space Agency (ESA) within their freely-distributed 'enhanced' data file, and generated using their Processing Baseline 2.27. Firstly, an estimate of the waveform noise was made from the mean power of the lowest six waveform samples, and waveforms where this value exceeded 0.3 of the maximum recorded power were rejected and not passed to the subsequent processing. Next, each remaining waveform was oversampled by a factor of 100 using a spline interpolation and then the leading edge of each waveform was identified based upon the first set of waveform samples that satisfied the following criteria, (1) a normalised power that exceeded the noise floor (defined as being 0.05 above the mean normalised power of the lowest 6 samples), (2) a change in normalised power from the noise floor to the next waveform peak that was greater than 0.2, and (3) an increase in power that ended with a defined waveform peak (such that there was a decrease in power at delay times beyond the peak location). Each waveform that had a leading edge satisfying these criteria was then retracked using several empirical retrackers, namely a Threshold on the offset Centre Of Gravity amplitude (TCOG) (*Wingham et al., 1986*), a Threshold First Maximum Retracker (TFMRA) (*Helm et al., 2014*), and a maximum gradient of the first leading edge retracker (*Gray et al., 2015*). For the first two solutions, a threshold of 50 % of the leading edge power (defined as the leading edge maximum power minus the noise floor) was used, with the aim of providing a stable retracking point across both low slope and more complex topographic surfaces. More specifically, we chose this mid-power threshold as a balance between minimising the sensitivity to noise at the start of the waveform leading edge, and reducing the impact of radar speckle (which is more apparent near to the waveform peak due to its multiplicative nature). For the majority of this study, we have focused on reporting results produced by the TCOG retracking, because of the continuity it provides with the ground segments of past European Space Agency (ESA) missions, and the broadly consistent results between all three of the retrackers tested. However, for completeness we do also report statistics from all retrackers within our independent validation exercise.

After retracking, Level-2 instrument and geophysical corrections were applied to each range measurement to account for the distance between the antenna and satellite centre of mass, dry and wet troposphere delays, ionosphere delays, solid Earth tide, ocean loading tide and Polar tide, plus ocean tide and the inverse barometer effect over floating ice. These corrections are all included within the enhanced data product, and further details can be found within the product specification (*Sentinel-3 Core PDGS Instrument Processing Facility (IPF) Implementation Product Data Format Specification, 2015*). These geophysical corrections are provided at 1 Hz sampling, and so we used linear interpolation to resample these fields to the native 20 Hz rate of the altimeter measurements. The echoing point was then relocated to the point of closest approach (*Roemer et al., 2007*) within the SAR beam footprint using a DEM derived from 7 years of CryoSat-2 data (*Slater et al., 2018*), with echoing points

that were relocated by more than ~ 8 km, and therefore at the edge of antenna beamwidth, removed (~ 3.7 % of data, based on statistics from a single cycle), together with relocated elevations that deviated by more than 100 metres from the DEM (~ 3.0 % of data, based on statistics from a single cycle). This dataset then formed the basis of the subsequent validation activities.

## 4. Delay-Doppler Measurement Precision

To assess the utility of the Sentinel-3A altimeter over ice surfaces, we firstly investigated the precision of the SRAL measurements, by assessing their repeatability in space and time. For this purpose we performed two sets of analysis, (1) an evaluation of repeated profiles that crossed subglacial Lake Vostok, a site that provides a stable and low-slope surface that is well established for validation studies *(Richter et al., 2014; Schröder et al., 2017; Shuman et al., 2006)*, and (2) a continent-wide single-cycle cross-over analysis, to evaluate the repeatability of measurements at locations where ascending and
descending satellite passes intersect (*Wingham et al., 1998; Zwally et al., 1989*).

### 4.1 Shot-to-shot Precision at Lake Vostok

We analysed repeated altimeter profiles that crossed the ice surface above the Lake Vostok site in East Antarctica, in order to assess the SRAL instrument precision. The smooth, flat surface above the lake minimises the influence of topography, and allowed us to focus primarily on the performance of the SRAL instrument itself, and specifically to understand the impact of
radar speckle, small-scale variations in the firn backscattering properties and the influence of retracker imprecision on the SAR altimeter measurements. Between December 2016 and December 2017, the satellite made 14 passes over the lake, and so, focusing on two ground tracks that crossed the flat ($< 0.01°$) central portion of the lake (Figure 2), we assessed the repeatability of these measurements in space and time. For each ground track, we used the 14 repeated elevation profiles to compute (1) the mean elevation profile, (2) the residual elevations from the mean profile, and (3) the standard deviations of all elevation
measurements within 400 m intervals along-track (Figure 2). Together these provide an assessment of the instrument shot-to-shot precision over ice sheet surfaces under the influence of minimal topography, and also the repeatability of measurements through time. We find that, over the first year of routine operations, the SAR altimeter has operated with sub-decimetre precision. On average, the 14-cycle 1-sigma standard deviation along both tracks was 7 cm, and rarely fell outside of the range of 5-10 cm along the entirety of the track segments analysed (Figure 2).

### 4.2 Single-cycle Cross-over Analysis

Next, we used a single-cycle cross-over analysis to assess the repeatability of measurements at all locations where ascending and descending satellite passes crossed. Elevation differences at orbital cross-overs are commonly used as a metric for measurement precision, and integrate a number of factors, including spatially uncorrelated orbit errors, retracker imprecision,

the impact of radar speckle, echo relocation errors and any sensitivity to anisotropic scattering within the near surface snowpack (*Armitage et al., 2014; Remy et al., 2012; Wingham et al., 1998*). We selected two 27-day cycles of Sentinel-3A data (cycles 12 and 24) and for each cycle computed the elevation difference at all locations where ascending and descending passes crossed. Specifically, we (1) removed outlier measurements that deviated by more than 50 metres from our DEM (*Slater et al., 2018*), (2) identified crossing points in the remaining dataset as the intersection between two consecutive measurements of an ascending pass and two consecutive measurements of a descending pass, and (3) computed the elevation difference by interpolating the bracketing ascending and descending records to the crossover location. Finally, for each cycle we binned all the cross-over differences within $0.2°$ intervals of surface slope, to investigate the relationship between the magnitude of the surface slope and the cross-over elevation precision. The results of this analysis are shown in Figure 3 and Table 2.

In total, approximately 90 000 cross-overs were identified during each cycle. At the continental-scale, the distribution of these elevation differences is non-normal (0.1% significance level). Cross-overs from both cycles have a median difference of < 1 cm in magnitude, and a higher than normal proportion of the differences are clustered around this central value, reflecting the good repeatability of measurements across the low slope interior of the ice sheet. For example, across the entirety of the ice sheet, the median absolute deviation of cross-overs from the median value is ~ 0.3 m, and 80% of all cross-over differences are less than 1.8 metres. At the extremes of these distributions, however, there are a number of outliers, with around 5% of cross-overs having an elevation difference that exceeds 10 metres. Generally, the magnitude of the cross-over differences increases with surface slope (Figure 3d), with the largest differences occurring in regions with steep and complex coastal topography. In these regions, the processes of locating the echoing point within the beam footprint, and of retracking complex multi-peaked waveforms become more challenging. These remain active topics of research, which are likely to deliver further improvements to ice sheet SAR altimetry in the future. In the meantime, we note that for many glaciological applications it may be beneficial to remove these outliers and, although not the focus of this study, we touch upon possible filtering strategies within Section 5.3.

## 5. Delay-Doppler Elevation Accuracy

### 5.1 Reference dataset and methods

To conduct an independent evaluation of the accuracy of our Sentinel-3A ice sheet measurements, we used elevation data acquired by the Airborne Topographic Mapper (ATM) and Riegl Laser Altimeter (RLA) instruments carried on Operation IceBridge campaigns flown between 2009 and 2016. At our two inland study sites we used the Level-2 ATM product, which provides surface elevation measurements with an along-track sampling every 0.25 seconds (equating to a ~ 30-meter interval for a typical aircraft velocity), and an 80-meter across-track platelet at nadir. The Level-2 product was chosen because the processing includes a smoothing of the Level-1b data, which reduces the impact of uncorrelated shot-to-shot noise on our

validation dataset, and also brings the measurement cell closer to the resolution of the SAR footprint. The ATM measurements have been estimated to have a vertical accuracy and precision of 7 cm and 3 cm, respectively (*Martin et al., 2012*). At our two coastal sites, where ATM measurements have not been acquired, we instead used RLA acquisitions. This instrument has a smaller ground footprint of 25 m along track by 1 meter across track, and a slightly larger reported accuracy of 12 cm

(*Blankenship et al., 2013*).

To compute elevation differences between our Sentinel-3A and IceBridge datasets, we identified IceBridge records within a 200-meter search radius of each satellite measurement. Where multiple IceBridge records existed within the search radius, we selected the closest measurement. Alternative methods for selecting IceBridge measurements were also tested, such as bilinear

interpolation of multiple surrounding measurements, but this approach produced a less comprehensive set of comparison points from which to generate our assessment statistics. As part of this process, we identified and removed anomalous IceBridge elevation records that deviated by more than 100 meters from an independent DEM (*Slater et al., 2018*). This step removed 7.3% of the total Antarctic IceBridge dataset. We then corrected for elevation differences arising from the spatial and temporal separation of the satellite and airborne measurements. In the case of the former, we constructed a bicubic interpolation of the

surrounding 4x4 pixel area of the DEM surface, and used this to estimate the difference in elevation between the satellite and airborne measurement locations. For the latter, we used an estimate of the local rate of elevation change (*McMillan et al., 2014*). The magnitude of the elevation change correction is small ($< 1$ cm yr$^{-1}$) at the Vostok, Dome C and Dronning Maud Land sites. At our Wilkes Land site, the magnitude of the correction is larger (8 cm yr$^{-1}$) and it is therefore possible that inaccuracies in the correction could contribute, in part, to the differences between the airborne and satellite measurements at

this site. For example, a 10 % error in the correction over a 5-year period would equate to a 4 cm error in the corrected IceBridge elevation. Finally, we computed the Sentinel-3 minus IceBridge elevation difference for each measurement pair, and so generated a set of statistics for each study site (Table 3). Because the differences, particularly at coastal sites, are not normally distributed, and exhibit higher clustering around the central value, together with a greater proportion of outliers, we principally use the median and median absolute deviation (MAD) from the median as measures of the bias and dispersion,

respectively. We choose to use the MAD, because for non-normal distributions this statistic provides a more representative measure of the mid-point of the dispersion. We do, nonetheless, report both the MAD and the standard deviation within Table 3.

## 5.2 Evaluation at Inland Sites

At the inland sites of Lake Vostok and Dome C, we find very good agreement between the Sentinel-3A and airborne datasets

(Figure 4 and Table 3). At Lake Vostok, the median bias between our TCOG solution and the airborne data is 1 cm, and the MAD dispersion of the differences is 13 cm. At Dome C, the bias is larger (20 cm), but the dispersion of the differences is smaller (6 cm). The differing bias between the two sites is investigated in more detail in Section 5.4. Comparing the results

from the different retrackers, we find variations of approximately 10-30 cm in the median bias, which reflects differences in the algorithms used to select the retracking point on the waveform leading edge. There is nonetheless relatively little difference between retrackers in the MAD of the elevation differences, which is typically of the order of 10 cm in magnitude (Table 3). This is consistent with our previous analysis of the instrument precision above Lake Vostok, and suggests that at these

relatively low slope inland sites, uncorrelated sources of error, for example due to imprecision of the retracker, radar speckle, the process of measurement relocation, or small-scale variations in snowpack characteristics, have not significantly affected the SAR altimeter elevation measurements. In total, we find that more than 97 % of validation points (TCOG retracker) have an elevation difference of less than 50 cm (Table 3), and that 70 % (Vostok) and 49 % (Dome C) have a difference of less than 20 cm.

**5.3 Evaluation at Coastal Sites**

At the coastal sites of Dronning Maud Land and Wilkes Land the differences between the Sentinel-3A and airborne datasets are, as expected, more widely dispersed than at our inland study locations (Figure 4 and Table 3). At these sites, the more rugged coastal topography can produce complex waveforms, as energy is often returned from several distinct surfaces within the illuminated beam footprint. These factors represent well-established challenges for radar altimetry, both for retracking

algorithms and for the procedure of correctly locating the on-the-ground origin of the derived elevation measurement. SAR altimetry, due to its smaller ground footprint, has the potential to be less affected by these topographic influences, and indeed we find that the overall median biases relative to IceBridge remain small (Table 3), namely 0.03 m and 0.12 m at Dronning Maud Land and Wilkes Land, respectively (TCOG retracking). The magnitude of these biases is comparable to those found at our inland sites, suggesting that for a metric that is robust to outliers, no systematic bias is introduced as large scale topographic

complexity increases.

For our coastal sites, the dispersion of the elevation differences relative to IceBridge is larger, as indicated by the MAD values of 0.30 m and 0.74 m (TCOG retracking) at Dronning Maud Land and Wilkes Land, respectively (Table 3). Nonetheless, these first results demonstrate that even in these more challenging regions, the MAD precision of SAR elevation measurements is

well below 1 meter. At these sites, we find that ~ 60-85 % of the validated satellite elevation measurements (TCOG retracking) are within 1 meter of their airborne counterpart, and 92-98 % are within 10 meters (Table 3). As is evident from these statistics, and also the standard deviation of the differences (Table 3), there are a small number of outlying measurements that exhibit larger deviations from the airborne validation data. Given that the primary focus of this study is on assessing data quality, we have chosen not to remove these outliers, although we note that for many future applications it may be beneficial to implement

filtering procedures to do so. For example, when we consider an Antarctic-wide evaluation (see Section 5.4), we find that removing Sentinel-3A points that deviate by more than 10 m from our pre-existing DEM achieves a 70% reduction in the standard deviation of the Sentinel-3 minus IceBridge differences. Although a small proportion of outliers still remain even

after this filtering step (~1 % of data deviate by more than 20 m), either because the DEM fails to identify them as outliers or because the IceBridge data themselves are inaccurate, it is evident that a simple post-processing strategy such as this may be beneficial for many glaciological applications.

## 5.4 Influence of Surface Topography on Sentinel-3 Measurements

To better understand the variations between the Sentinel-3A and IceBridge measurements at our different sites, we considered the influence of both large-scale (wavelength much greater than the beam footprint, i.e. surface slope) and small-scale (wavelength equal to, or shorter than, the beam footprint) topography. To investigate the former, we expanded our site-specific analysis to calculate elevation differences between one complete cycle of Sentinel-3A data (cycle 12, elevations derived using the TCOG retracker) and all ATM and RLA measurements acquired across the Antarctic ice sheet between 2009 and 2016, in

order to assess the sensitivity of the elevation differences to surface slope (Figure 5). At the continent scale we find a Sentinel-3A minus IceBridge median elevation difference of 0.06 m, and a MAD of the elevation differences of 1.06 m. It is important to note that these statistics do not represent an unbiased sample of the total ice sheet distribution. The IceBridge surveys have more frequently been flown across steep and more complex ice margin topography (Figure 5b), resulting in a median slope at the locations where the Sentinel-3A data is validated by IceBridge that is 37% higher than the median slope of the ice sheet as

a whole.

Nonetheless, the IceBridge dataset covers a range of surface slopes, and allows us to investigate the relationship between the magnitude of the surface slope and the Sentinel-3A – IceBridge elevation differences. Specifically, we grouped the validation data within 0.2° slope intervals (using the slope model presented in *Slater et al., 2018*), and computed the median absolute

elevation difference across each band (Figure 5). Unsurprisingly, the magnitude of the median elevation difference increases with the magnitude of the surface slope, mirroring the trend in precision that was apparent in our cross-over data, and reflecting the challenges of retracking and reliably locating the echoing point in areas of steeply sloping terrain. For surface slopes below 0.4°, which constitutes 74% of the ice sheet area, the median absolute difference between Sentinel-3 and IceBridge is less than 1 metre, whereas for highly sloping terrain greater than 1° the difference increases to ~ 10 metres. We note that these statistics

are specifically related to the current processing baseline, and that expected improvements to the Sentinel-3 processing chain are likely to improve performance in highly sloping areas in the future (see further discussion below).

Considering the validation statistics across all four study sites, the pattern of increasing dispersion of elevation differences at coastal locations is consistent with our understanding that measurement precision degrades with increasing topographic

complexity. In contrast, it was less expected that, for any given retracker, we find a difference of ~ 20 cm in the bias recorded at the Lake Vostok and Dome C sites, especially because the large-scale topographic characteristics (surface slope < 0.1°) and climatological setting (cold, dry ice sheet interior) are similar. Although an inaccurate elevation rate correction could introduce

a bias into our measurements (due to the difference in date between the Sentinel-3A and IceBridge acquisitions) the median elevation rates at these inland sites are small (< 1 cm yr$^{-1}$ at both Dome C and Vostok) and so we believe it is unlikely that this is the source of the observed difference. To investigate other possible physical explanations for this difference, we therefore used the IceBridge data to assess the finer scale topography at both of these inland sites. Specifically, we estimated the long wavelength signal along each elevation profile by fitting a quadratic curve to the data, and then plotted the elevation residuals having removed the modelled long-wavelength topography (Figure 6). For this analysis, we focused specifically on the inland sites because (1) they exhibited a differing, and unexplained, bias relative to IceBridge, and (2) at long-wavelengths they are relatively flat, and so any impact of small-scale roughness is likely to be more evident than at coastal sites with more complex long-wavelength topography. The airborne flightlines (Figure 6) show that Dome C presents a much rougher surface at 100-500 m length scales, with amplitudes typically ranging from ~ 5-30 cm. In comparison, the amplitude of oscillations at Lake Vostok is much smaller, typically 1-5 cm. Along these profiles the standard deviation of the residuals is 1.7 cm and 6.9 cm for Lake Vostok and Dome C, respectively, indicating that by this metric of surface roughness, Dome C is ~ 4 times rougher than Lake Vostok. It is therefore possible that part of the larger bias at Dome C can be explained by the rougher surface, and the tendency of the satellite altimeter, given its larger footprint, to be more influenced by the local topographic peaks than the airborne instrument. Finally, it should be noted that any of the retracking algorithms can likely be tuned to reduce the bias at a particular site, for example by selecting a higher retracking point on the waveform leading edge that is closer to the theoretical mean return from a surface with these roughness characteristics. However, we reiterate that our philosophy here is to use a conservative retracking threshold that is likely to deliver robust and stable results across all types of topographic regimes, and one that is therefore well-suited to delivering reliable continent-wide estimates of surface elevation change through time.

## 6. Ice Sheet elevation change from Delay-Doppler Altimetry

Our analysis has provided the first comprehensive assessment of ice sheet elevation measurements that have been derived using Sentinel-3A Delay-Doppler (SAR mode) altimetry, and an initial demonstration of their accuracy and precision across a range of topographic regimes. Within a wider geophysical context, one of the principle uses of altimetry data is to determine changes in ice sheet elevation over time (*Flament & Rémy, 2012; Shepherd et al., 2012; Shepherd & Wingham, 2007; Zwally et al., 2005*). Although the available time span of Sentinel-3A acquisitions is short for detailed glaciological interpretation of any signals, it is nonetheless of interest to establish (1) the extent to which the system has provided a stable measurement platform in time, and (2) whether the precision and accuracy of the SAR mode of operation is sufficient to be able to resolve known signals and modes of glaciological change. As a preliminary investigation of these questions, we therefore applied a modified model-fit method (*McMillan et al., 2014, 2016*) to all Sentinel-3 data acquired up to and including cycle 32, in order to explore the potential of these data for mapping elevation changes of the Antarctic Ice Sheet. In summary, we firstly removed an *a priori* estimate of elevation from each measurement using an auxiliary DEM (*Slater et al., 2018*), rejecting records that deviated by more than 50 m from the same DEM. We then used the resulting elevation residuals to simultaneously solve for

linear spatial and temporal rates of elevation change on a 5 x 5 km grid. We rejected grid cells where the model produced a poor or geophysically unrealistic fit to the data, defined to be where the root-mean-square of the observed-minus-modelled residuals exceeded 2 meters, the absolute rate of elevation change exceeded 10 m yr$^{-1}$, the spatial gradient computed from the elevation residuals exceeded 5$^o$, or where less than 20 measurements constrained the model fit.

Using this method, we determined an estimate of the rate of ice sheet surface elevation change across a total area of 5 061 700 km$^2$, constituting 42.3 % of the ice sheet (Figure 7). Across large parts of the slow-flowing ice sheet interior, the derived rates of elevation change are low. This agrees with numerous recent studies (*Flament & Rémy, 2012; Helm et al., 2014; McMillan et al., 2014*), and provides an early indication that the Sentinel-3 instrument and orbital configuration is suitable for mapping changes across the low relief ice sheet interior. Although we believe that the Sentinel-3 record is still too short to perform a detailed, ice sheet-wide, quantitative inter-comparison relative to previously published altimeter datasets, we do find evidence that Delay-Doppler altimetry is able to map the higher, dynamically-driven, rates of elevation change that are occurring across coastal regions of the ice sheet (*Flament & Rémy, 2012; Helm et al., 2014; McMillan et al., 2014*). In particular, we observe widespread elevation change across the fast-flowing ice streams draining into the Amundsen Sea Sector of West Antarctica, with rates of surface lowering of 2-3 m yr$^{-1}$ close to the grounding line of Pine Island Glacier and 4-5 m yr$^{-1}$ upstream of the grounding lines of the Thwaites and Smith Glaciers. In comparison to inland regions, the coverage is generally poorer across the steeper ice margin regions and the more mountainous terrain of the Antarctic Peninsula. This is expected given the lower precision of elevation measurements in these locations, the wider track spacing, and the short time period over which the trends are being computed. Additionally, it is important to note that the current approach to performing the waveform windowing and SAR multi-looking within the Level-1B processing chain of the ground segment is not fully optimized for these challenging ice regions. Further refinements to this processing step, namely to adjust the windowing during the Doppler beam stacking to account for large variations in the satellite tracker range, are currently being implemented, and are expected to deliver future improvements in data retrieval in these regions. Based upon this preliminary assessment, however, there is good reason to expect that Sentinel-3 Delay-Doppler altimetry will prove to be an effective tool for mapping ice sheet elevation change.

Finally, we investigated the capability of SAR altimetry to make precise measurements of surface elevation change within the inland regions of the ice sheet, where previously only Low Resolution Mode observations have been available. Specifically, we focused on a small region (of the order of tens of kilometres) of anomalously high elevation changes at a location within the interior of the East Antarctic Ice Sheet (location shown in Figure 7a). We analysed 28 cycles of Sentinel-3A data passing over this region, grouping data from all cycles within 340 m intervals along-track, and again employing a model fit approach to isolate the temporal evolution of the ice surface with a 27-day repeat frequency (*Moholdt et al., 2010; Smith et al., 2009*). On either side of this region of high elevation change, we find a high level of repeatability of the SAR elevation measurements, giving us confidence in the precision of the instrument and our method for isolating along-track elevation changes through

time (Figure 7b). Over the feature itself, we resolve a progressive, spatially coherent movement of the ice sheet surface, with a total lowering of ~ 1.7 metres over a period of 16 months. Transient changes in elevation at this spatial scale are widely understood to be caused by subglacial lake drainage (*Smith et al., 2009*), and as such, our observations provide a first indication of the capability of Sentinel-3 SAR altimetry to systematically monitor such events.

## 7. Conclusions

We have undertaken a first assessment of the utility of Sentinel-3 Delay-Doppler (SAR mode) altimetry for measuring ice sheet elevation and elevation change using the standard ESA Level-1b product and our own Level-2 processing chain. Analysis of repeated acquisitions over the Lake Vostok validation site indicates that, over the first year of routine operations, the instrument has operated with sub-decimeter precision. Through validation with airborne campaigns, we find small median biases in elevation, typically of the order 1-10 cm, at both inland and coastal sites. The dispersion of elevation residuals, measured with respect to the validation data, is of the order of 10 cm at inland sites, increasing to ~1 m at coastal sites with more complex topography. This reflects the main challenges associated with processing radar altimetry data in complex ice margin regions, namely (1) reliable retracking of multipeak waveforms that arise when multiple distinct surface reflections are captured within the receive window, and (2) accurately establishing the location of the echoing point within the SAR beam footprint. These represent principle avenues of future research within the field of ice sheet Delay-Doppler altimetry. Nonetheless, the accuracy achieved in even these complex ice margin regions is encouraging, and expected to improve further as refinements are made to the operational ground segment processing. Finally, we have shown the capability of Sentinel-3, albeit with the relatively short record of data currently available, to resolve the known signals of elevation change that currently dominate Antarctica's contribution to sea level rise, and to monitor subglacial lake activity. Together, our analysis demonstrates the early promise of Sentinel-3 SAR altimetry as a platform for long-term, operational monitoring of Earth's ice sheets.

## 8. Data Availability

The Sentinel-3A altimetry data used in this study are freely available through the Copernicus Open Access Hub (https://scihub.copernicus.eu/dhus/#/home). The IceBridge airborne altimetry data used in this study are freely available from the US National Snow and Ice Data Center (https://nsidc.org/). The CryoSat-2 DEM used in this study is freely distributed by the UK NERC Centre for Polar Observation and Modelling (http://www.cpom.ucl.ac.uk/csopr/icesheets2).

## 9. Author Contribution

MM designed the experiments. MM and AM processed and analysed the data. MM prepared the manuscript with contributions from AS, MR, AA and JB, and all authors reviewed the manuscript. The authors declare that they have no conflict of interest.

## 10. Acknowledgements

This work was supported by the UK NERC Centre for Polar Observation and Modelling, the European Space Agency contract *SEOM – Sentinel-3 Performance Improvements for ICE sheets* (contract number 4000115201/15/I-BG), and the *Sentinel-3 Mission Performance Centre*. We thank the Editor G. Catannia and two anonymous reviewers for their comments, which have substantially improved the manuscript.

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

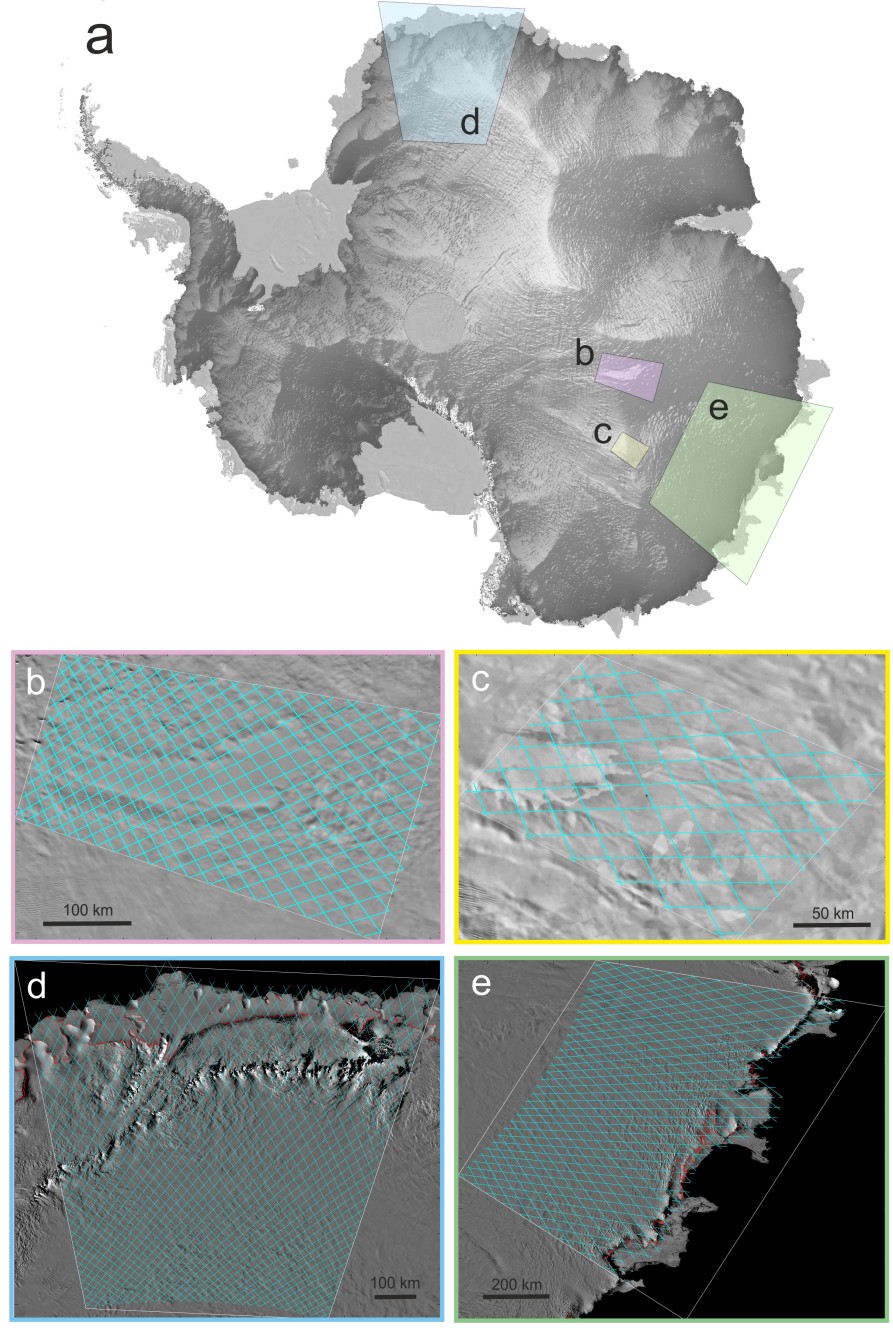

**Figure 1. a. Overview of the Lake Vostok (b), Dome C (c), Dronning Maud Land (d) and Wilkes Land (e) study sites. The background image in panel a is a surface DEM from CryoSat-2 (*Slater et al., 2018*) overlaid upon the MODIS Mosaic of Antarctica (MOA) (*Haran et al., 2006*). Panels b-e show the Sentinel-3 ground tracks (cyan) at each study site, overlaid upon MOA.**

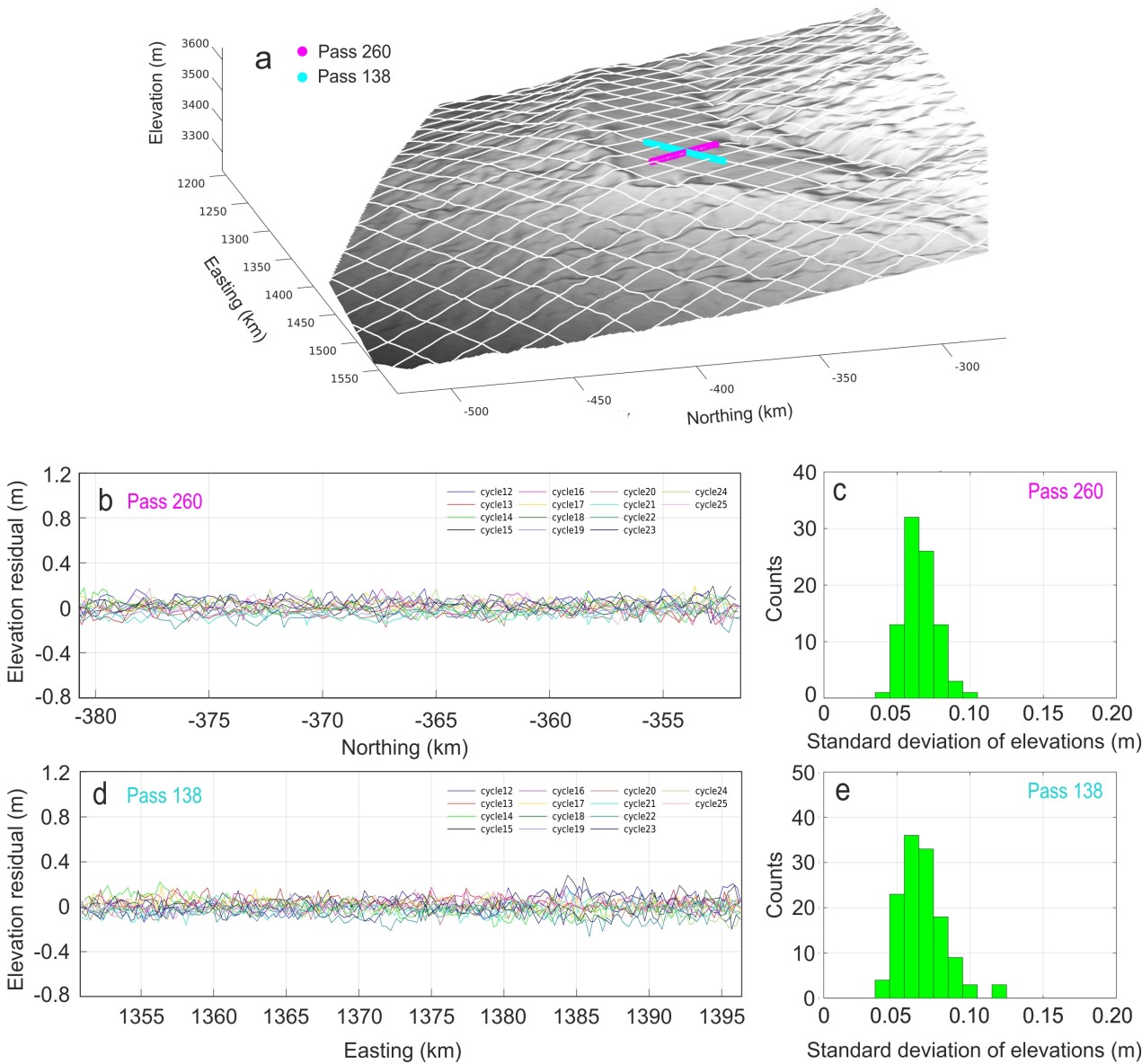

**Figure 2. Assessment of instrument precision at the Lake Vostok site in East Antarctica. a. The location of the two ground tracks crossing the center of the lake (Pass 138, cyan; Pass 260, magenta), plotted on a surface DEM (*Slater et al., 2018*), with other passes shown in white. For each pass, 14 cycles were accumulated between Dec. 2016 and Dec. 2017. Panels b and d show the residuals from the mean elevation of all cycles. Panels c and e show the distributions of the standard deviation of elevation in each 400 m intervals along the satellite track.**

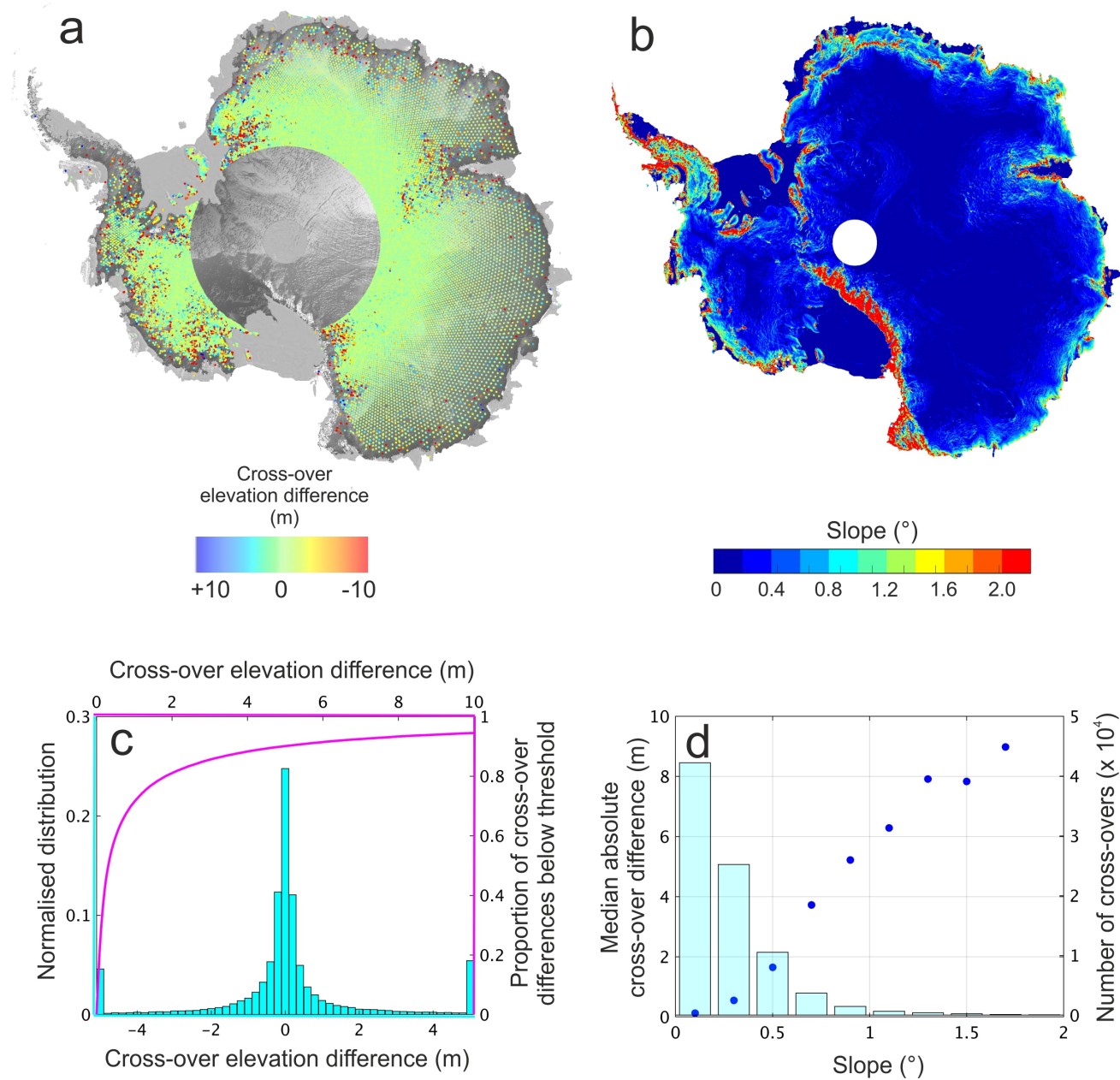

**Figure 3. Elevation differences at orbital cross-overs for cycle 12 of the Sentinel-3A mission, and comparison to the gradient of the surface slope. a. Cross-over elevation differences, b. Surface slope from an independent DEM (*Slater et al., 2018*), c. The distribution of cross-over elevation differences (cyan histograms and axes), and the cumulative distribution of the absolute elevation differences (magenta curves and axes), and d. the median absolute cross-over elevation difference (blue dots) and number of cross-overs (blue bars) as a function of surface slope.**

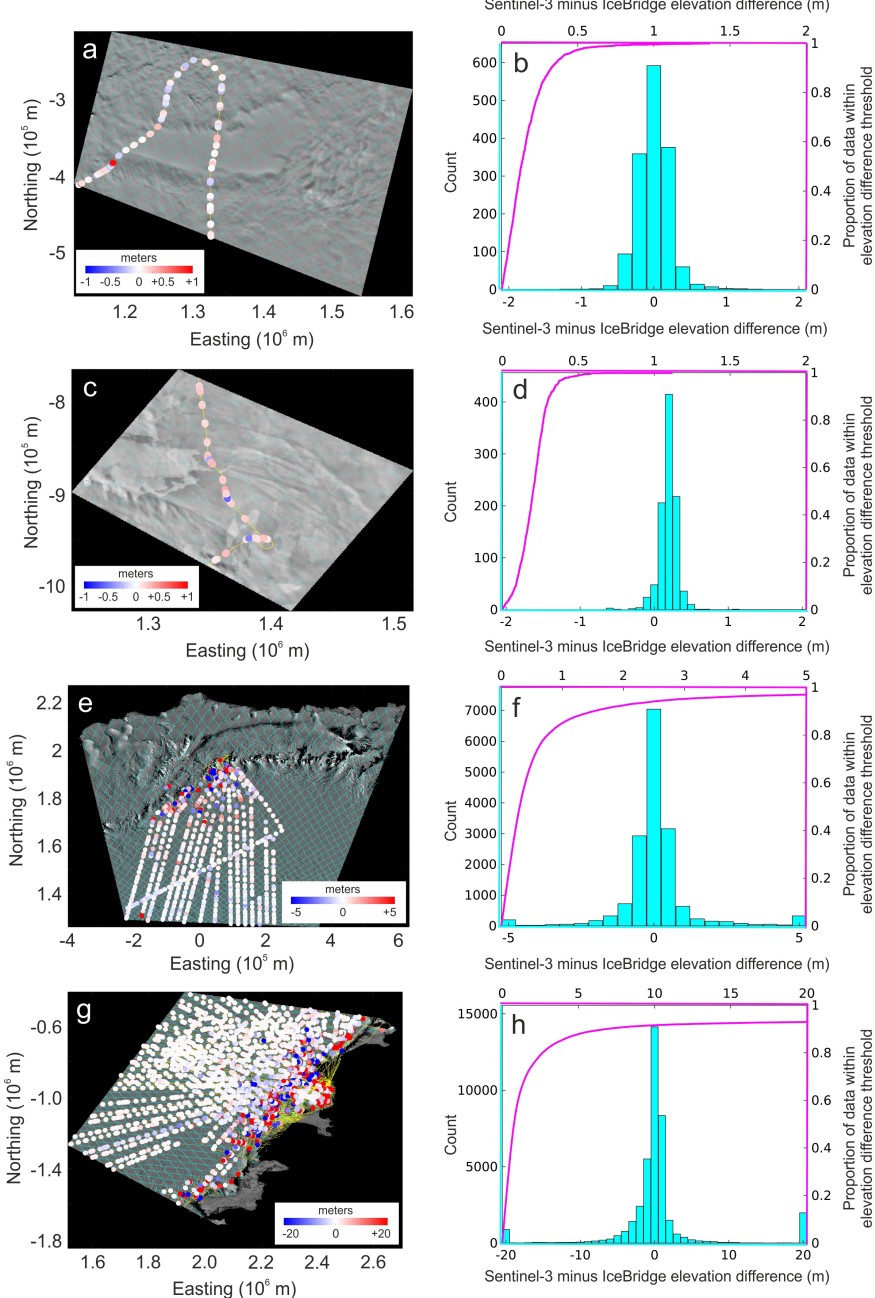

**Figure 4. Assessment of the accuracy of Sentinel-3A elevation measurements at the Lake Vostok (a,b), Dome C (c,d), Dronning Maud Land (e,f) and Wilkes Land (g,h) sites in East Antarctica. a,c,e,g. Elevation differences between Sentinel-3 and IceBridge, overlaid upon the MOA (*Haran et al., 2006*), with the Sentinel-3 tracks shown in cyan. b,d,f,h. The distribution of Sentinel-3 minus IceBridge elevation differences (cyan histograms and axes), and the cumulative distribution of the absolute Sentinel-3 minus IceBridge elevation differences (magenta curves and axes) at each site.**

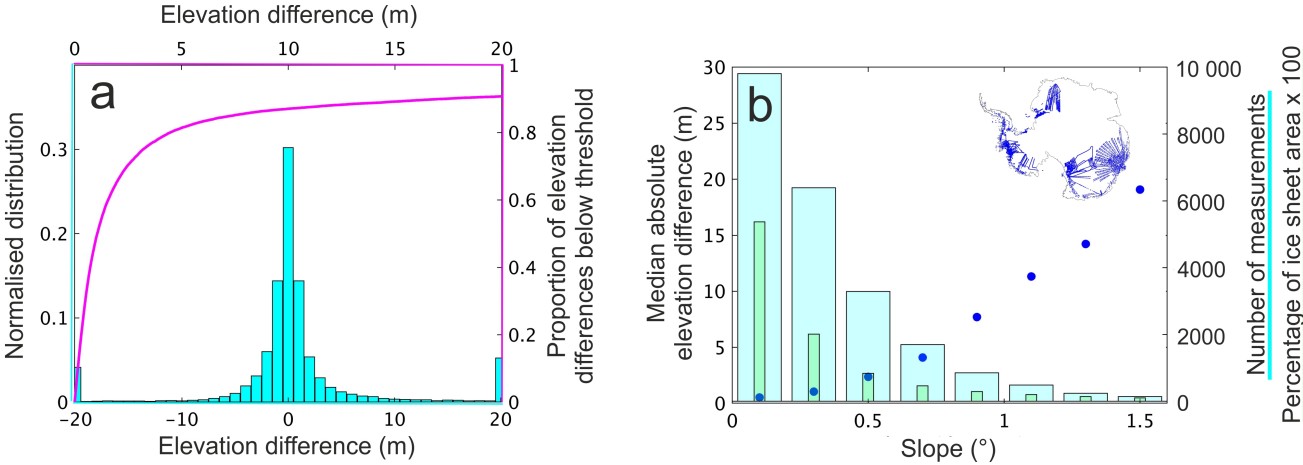

**Figure 5. Ice sheet wide assessment of the accuracy of Sentinel-3A elevation measurements acquired during cycle 12. a. The distribution of Sentinel-3 minus IceBridge elevation differences (cyan histograms and axes), and the cumulative distribution of the absolute Sentinel-3 minus IceBridge elevation differences (magenta curve and axes). b. the median absolute Sentinel-3 minus IceBridge elevation difference (blue dots), the number of validation measurements (blue bars), and the percentage of the ice sheet (green bars) within 0.2° slope bands. The inset figure shows the spatial distribution of the IceBridge validation measurements.**

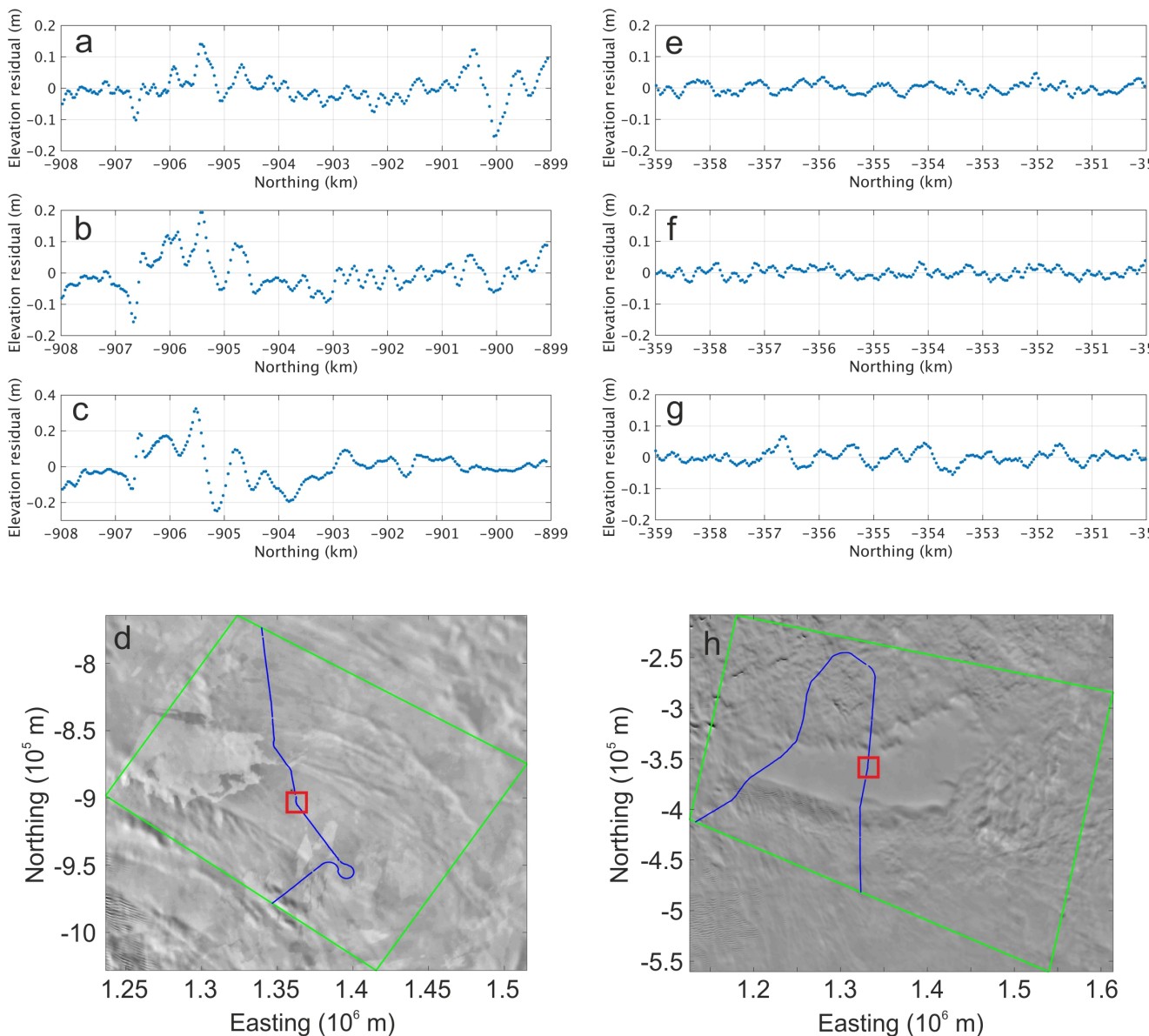

**Figure 6. Comparison of 100 meter scale surface roughness at Dome C (a-d) and Lake Vostok (e-h). Panels a-c (Dome C) and e-g (Vostok) show profiles of elevation residuals along each of the three sub-tracks resolved within the ATM instrument swath. The locations of the airborne flight lines are marked by the red boxes in panels d and h, respectively. Residuals are computed by removing a quadratic trend from each elevation profile. In panels d and h, the airborne ground tracks are shown in blue, the bounds of the study area in green, and the background image is from the MODIS Mosaic of Antarctica (MOA) (*Haran et al., 2006*).**

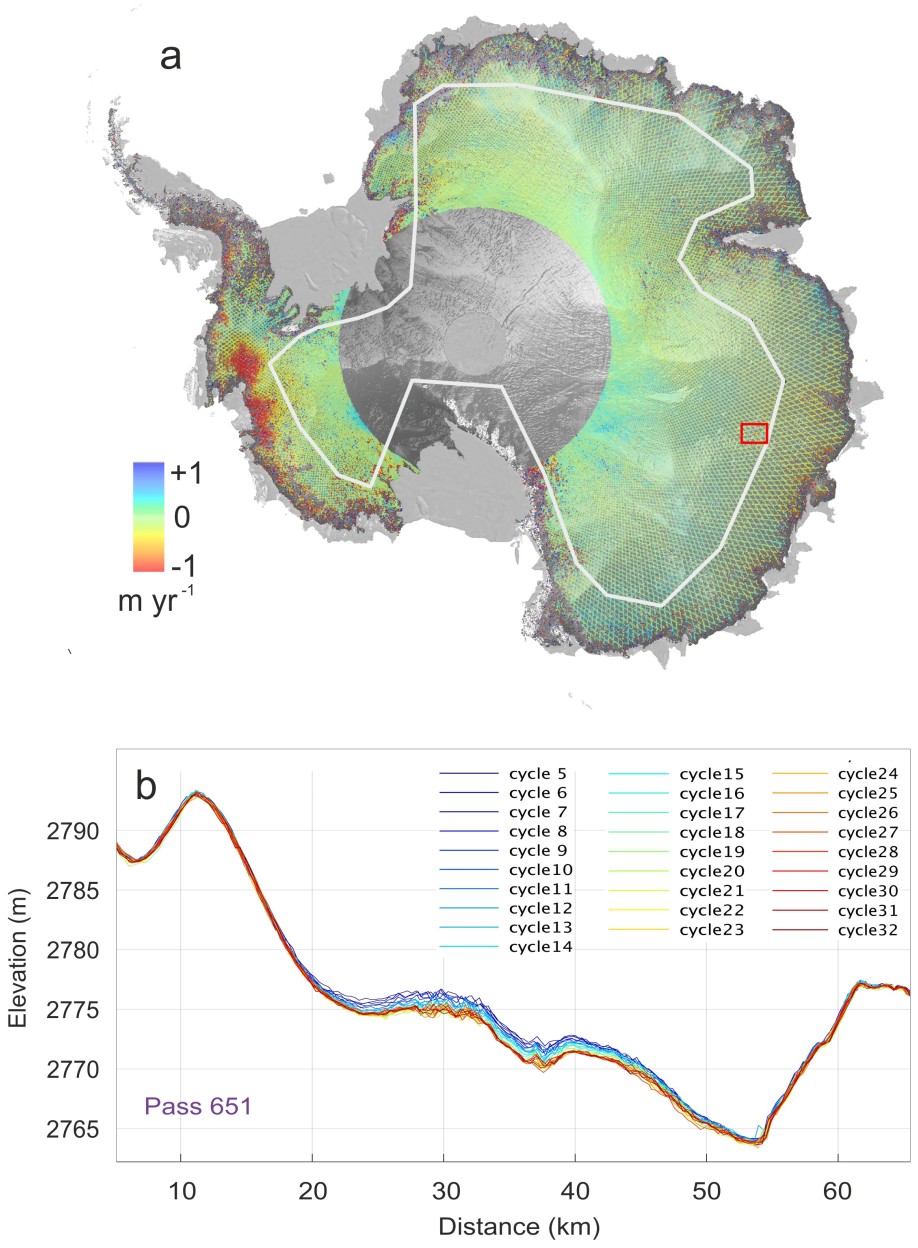

**Figure 7a. Rates of Antarctic surface elevation change derived from Sentinel-3A Delay-Doppler altimetry acquired between May 2016 and June 2018. The background image is a shaded relief derived from a DEM (*Slater et al., 2018*), overlaid upon the MODIS Mosaic of Antarctica (MOA) (*Haran et al., 2006*). The white line shows, for reference, the CryoSat-2 mode mask boundary, which separates regions where CryoSat-2 operates in Low Resolution Mode (interior regions) and in SAR interferometric mode (coastal regions). The absence of Sentinel-3 elevation rate data in the region surrounding the South Pole is due to the 81.35° latitudinal limit of the satellite orbit. b. Localised surface lowering resolved by Sentinel-3A across a 30 km track segment in East Antarctica (location shown by red box in panel a), indicating a possible subglacial lake drainage event.**

**Table 1. Summary of Sentinel-3 orbit characteristics and primary altimeter payload.**

| | |
|---|---|
| Sentinel-3A launch date | 16[th] Feb. 2016 |
| Sentinel-3B launch date | 25[th] Apr. 2018 |
| Sentinel-3C planned launch date | 2023 |
| Sentinel-3D planned launch date | 2025 |
| Orbital Inclination | 98.65° |
| Orbital altitude | ~ 830 km |
| Repeat period | 27 days |
| SRAL Central Frequency | 13.575 GHz |
| Antenna beam-limited footprint diameter | ~ 18 km |
| Along-track measurement interval | ~ 330 m |
| Along-track resolution | ~ 300 m |
| Across-track resolution | ~ 1600-3000 m |

**Table 2. Sentinel-3A single-cycle cross-over statistics.**

|  | Cycle 12 | Cycle 24 |
|---|---|---|
| Number of cross-overs | 85 100 | 88 757 |
| Median cross-over difference (m) | -0.004 | 0.001 |
| Mean cross-over difference (m) | 0.19 | 0.20 |
| Median absolute deviation of elevation differences (m) | 0.31 | 0.28 |
| Standard deviation of cross-over differences (m) | 7.06 | 7.02 |
| RMS of cross-over differences (m) | 7.07 | 7.03 |

**Table 3.** Sentinel-3A validation statistics based upon comparison to IceBridge airborne altimetry at the four study sites. Results are given for elevations derived using a Threshold on the offset Centre Of Gravity (TCOG), Threshold First Maximum Retracker (TFMRA), and maximum gradient of the first leading edge (Max. Grad.) retracker. Elevation differences are calculated as Sentinel-3A elevation minus IceBridge elevation.

| | Number of measurements | Median elevation difference (m) | Mean elevation difference (m) | Median absolute deviation of elevation differences (m) | Standard deviation of elevation differences (m) | RMS of elevation differences (m) | % points within 0.5 metres of IceBridge elevation | % points within 1 metre of IceBridge elevation | % points within 10 metres of IceBridge elevation |
|---|---|---|---|---|---|---|---|---|---|
| **Vostok[1]** | | | | | | | | | |
| TCOG | 1523 | 0.01 | -0.001 | 0.13 | 0.22 | 0.22 | 97.3 | 99.5 | 100.0 |
| TFMRA | 1523 | -0.15 | -0.16 | 0.13 | 0.23 | 0.27 | 93.8 | 99.6 | 100.0 |
| Max. Grad. | 1523 | -0.25 | -0.28 | 0.16 | 0.33 | 0.43 | 83.5 | 97.7 | 100.0 |
| | | | | | | | | | |
| **Dome C[2]** | | | | | | | | | |
| TCOG | 971 | 0.20 | 0.19 | 0.06 | 0.12 | 0.23 | 98.9 | 99.9 | 100.0 |
| TFMRA | 971 | 0.06 | 0.05 | 0.06 | 0.12 | 0.13 | 99.5 | 100 | 100.0 |
| Max. Grad. | 971 | -0.08 | -0.10 | 0.10 | 0.20 | 0.23 | 95.8 | 99.5 | 100.0 |
| | | | | | | | | | |
| **Dronning Maud Land[3]** | | | | | | | | | |
| TCOG | 16462 | 0.03 | 0.42 | 0.30 | 7.30 | 7.31 | 68.5 | 85.0 | 98.1 |
| TFMRA | 16538 | -0.11 | 0.31 | 0.31 | 7.39 | 7.40 | 66.3 | 84.4 | 98.0 |
| Max. Grad. | 16538 | -0.24 | 0.13 | 0.33 | 7.40 | 7.40 | 59.5 | 81.9 | 98.0 |
| | | | | | | | | | |
| **Wilkes Land[4]** | | | | | | | | | |
| TCOG | 40400 | 0.12 | 1.43 | 0.74 | 14.99 | 15.06 | 35.0 | 58.8 | 91.6 |
| TFMRA | 40803 | -0.03 | 1.28 | 0.75 | 14.94 | 14.99 | 37.3 | 58.5 | 91.6 |
| Max. Grad. | 40799 | -0.20 | 0.98 | 0.83 | 14.96 | 15.00 | 35.9 | 55.2 | 91.4 |

1. Vostok IceBridge measurements were acquired during the months of 10/2013 and 11/2013.
2. Dome C IceBridge measurements were acquired during the months of 10/2013 and 11/2013.
3. Dronning Maud Land IceBridge measurements were acquired during the month of 2/2011.
4. Wilkes Land IceBridge measurements were acquired during the months of 1/2009, 2/2009, 12/2009, 1/2010, 12/2010, 1/2011, 12/2011, 11/2012 and 12/2012.

