# Peer review of "Sentinel-3 Delay-Doppler Altimetry over Antarctica"

_The Cryosphere, 2018_

## Referee Comment (RC1) · Anonymous Referee #1 · 1 Aug 2018

This paper presents some of the first results from Sentinel 3's doppler focused radar altimeter over Antarctica. It describes the instrument and processing strategies, and it gives evidence that the data have decimeter precision over flat targets, and up to a few meters precision over sloping surfaces.

The paper is well written and needed very little editing. Its introduction to Sentinel-3 data should be valuable to the research community, and it makes the case well that the data are useful. The paper is somewhat weak in the analysis (rather than illustration) of the data.

I would have liked to see the elevation error magnitudes explained by mechanisms, rather than just quantified by region: it appears that even in the DML and Wilkes Land areas, the errors are multimodal, with a narrow distribution in flat areas and a wide

distribution in areas with high relief. Were I using the data, I would want to assign error estimates to the data based on, for example, the local surface slope or the footprint-scale relief. Calculating errors based on histograms for areas hundreds of km on a side does not do much to help me do this. Likewise, it would be useful if the authors could explore a quantitative connection between the roughness differences at Dome C and Vostok and the biases, and perhaps include the other sites in this analysis. Without understanding the roughness at the DML and Wilkes sites, I don't know if the roughness difference between Vostok and Dome C is significant.

My other concerns about the figures. Fonts are consistently too small, and lines are consistently too fine. The authors should print out their paper and try to read the figures!

Figure 2: Panels d and e show the same thing as b and c. I recommend just showing just one of these profiles. The green bars do not seem to show anything different from one part of the pass to another. I would recommend showing a histogram of the data contained in the bars (i.e. a histogram of the 400-m standard deviations).

Figures 3 and 4: It is hard to tell if the DEM spans the axes, or if there is extra white space around the DEM surface. The elevation axis is illegible, as are the labels on the inset panels.

Figure 3: The span of the colour scale for the coloured points is too large, and should be reduced to the span of the residuals.

Figure 5: I could not find a description of the differences between panels A-C, and between panels E-G, either in the text or the caption. Are there multiple profiles in d and h? The lines and points in D and H are too small, and the axis label font sizes are unreadable.

---

## Referee Comment (RC2) · Anonymous Referee #2 · 6 Sep 2018

**General comments**

This manuscript assesses the quality of Sentinel-3 delay/Doppler radar altimetry over Antarctica through analysis of: precision at Vostok Subglacial Lake and Dome C; accuracy at Vostok, Dome C, Dronning Maud Land, and Wilkes Land; and elevation change across the continent. The authors find that Sentinel-3 altimetry achieves a precision of $<0.1$ m over flat topography, has a mean bias $<0.2$ m in the flat interior and $<1.5$ m in complex coastal terrain, and can reproduce known patterns in elevation-change rate at the continent scale. The paper represents the first use of Sentinel-3 altimetry data over an ice sheet, is generally well-written, and will be of significant value to the community as an introduction to the capabilities of this Sentinel mission, but there are three major weaknesses that should be addressed before publication:

1. **Limited Analysis:** While the overall analysis presented in the manuscript is sound, it is also much more limited than I would expect from a rigorous treatment of altimetry uncertainty. The authors only assess precision in two flat, interior locations, yielding a "best-case" estimate, while opting not to perform a continent-wide crossover analysis, which would provide the readers with much better insight how and why precision varies. Crossover analysis is a standard technique in satellite altimetry error analysis and should be included in any rigorous assessment. In a similar vein, there are only four regions included in the accuracy analysis, when Operation IceBridge has surveyed across the continent. It is not clear why this assessment was so limited in scope, when including the rest of the Operation IceBridge airborne laser altimetry dataset would help illuminate the causes of error in the Sentinel-3 altimetry and provide the community with much more direction as to where and how to use this new dataset. Finally, the analysis of elevation-change rate is entirely qualitative; while I certainly understand that the processing chain is still in development, there should be some quantitative analysis of the result if it is to be included in the manuscript, such as a comparison to CryoSat-2 derived elevation-change rates over the same time window. At the very least, a map of "known" $dh/dt$ from CryoSat-2 or ICESat should be shown in conjuncture with the Sentinel-3 map to show the patterns are generally similar.

2. **Data and Method Detail:** There are quite a few pieces of information missing with respect to the data sources used and the methods applied. There is a long list of corrections applied to the altimetry data, but no citations that provide the details of the models. There is no data citation related to the ATM laser altimetry (though there is for the Riegl data), which prevents the reader from knowing what level of data processing was used or other important details for understanding cm-scale accuracy. It appears to be L2 data; why was L2 chosen over L1b (full-swath)? Are the same corrections with the same models applied to the airborne

altimetry as to the satellite altimetry? What are the dates of the Operation Ice-Bridge flights (which is needed understanding the potential impact of the $dh/dt$ correction)? Last, the method for correcting space and time elevation differences for the satellite-airborne analysis requires some more detail: how is a 1 km DEM being used to correct for spatial separations of $<100$ m? What is the error associated with the $dh/dt$ correction? What is the magnitude of these corrections (mm? cm? m?)? Also, at a basic level, are the results presented as $h_{sat} - h_{air}$ or $h_{air} - h_{sat}$? The former is typical, but the manuscript implies the latter ("IceBridge-Sentinel-3 measurement pair"); resolving this issue is fundamental to the analysis of why the bias exists (p.7, lines 7–9).

3. **Figure Presentation:** The figures could be greatly improved to make them easier to understand and to use journal space better. Cyan and yellow for groundtracks on Figures 1, 3, and 5 are hard to see, printed or on a screen. I am not sure why the maps on Figures 2, 3, and 4 are plotted as 3D perspective plots; this style makes it harder to understand the spatial context, particularly how the tracks align with easting and northing (which is important since this is how the transects are plotted). The axis and legend labels on all figures are unreadably small and the line widths are too small. I am not convinced that Figures 2c and 2e add to the discussion over a single histogram of uncertainty in 400 m along-track bins across the entire elevation range. The histograms and cumulative distribution functions in Figures 3b/3c and 3e/3f can likely be combined into a single panel. The authors should rethink how the each of the figures is presented to make the best use of space and to ensure they are fully legible at production size.

**Specific comments**

- p.1, line 13: "the first step towards a new era" is a bit over the top. The Antarctic coast is already blanketed in *better* altimetry (that includes along-track delay/Doppler processing). Please tone down these statements throughout (especially the use of "novel").

- p.1, line 20: "accuracy decreases slightly" is an understatement, given most of the measurements are up-to an order of magnitude less accurate.

- p.2, line 16: the rest of the details of what controls pulse-limited footprint should go in this parenthetical statement (i.e., satellite altitude, pulse length).

- p.3, lines 5–7: Figure 1 should have the SARIn/LRM mode mask on it to show where Sentinel-3 altimetry will make the biggest improvement compared to existing datasets (i.e., where Sentinel-3 covers LRM areas). It also should show the latitudinal limit of Sentinel-3.

- p.4, lines 12–14: The rest of the study *only* uses the TCOG retracker; I think adding some of the statistics from the other retrackers to Table 2 would be a fantastic addition and helpful for the community.

- p.4, lines 23–24: I would like to see the % of data points removed by this filtering step.

- p.4, line 27: This date range is insufficient given the $dh/dt$ correction. Please provide exact flight dates, either here or in Table 2.

- p.4, line 28: The along-track sampling of ATM L2 data is 0.25 s, which results in variable along-track distance spacing due to changes in aircraft velocity.

- p.5, line 3: Why was the single closest measurement chosen, instead of some sort of average of all the points within the approximate SAR footprint? Given that surface roughness is later suggested as the cause of bias and that lidar can easily resolve surface roughness, it seems appropriate to try a less simplistic approach that can account for surface roughness.

- p.5, lines 8–10: A few words about how the distributions deviate from normal would be appreciated here. I am also somewhat worried about the use of median absolute deviation given that it is >20x smaller than the standard deviation in coastal areas, demonstrating that outliers are a significant problem for Sentinel-3 altimetry. One path forward would be to suggest a filtering method for Sentinel-3 users could apply that would bring the two statistics closer to one-another.

- p.5, lines 3–4: What % of IceBridge records were removed by this filtering step?

- p.5, line 22: It should be noted that you are using $1\sigma$ as the definition of precision.

- p.6, lines 4–5: While it may be true, I do not think that the conclusion that the SAR waveform leading edge is insensitive to penetration is defensible given the analysis presented in the manuscript. Perhaps it is the Sentinel-3 sampling geometry that limits observed variability, or, even more simply, there was not much surface variability during the one-year period of observations. This statement should be removed, or alternatively there needs to be a thorough comparison between Sentinel-3 SAR and CryoSat-2 LRM mode using a similar retracker over the overlapping period to demonstrate that SAR processing (specifically) reduces sensitivity to penetration and subsurface scattering.

- p.6, lines 8–9: A "slope" correction, in its original meaning, is not applied in this processing chain; it is a "relocation" correction.

- p.6, lines 28–29: Given the importances of these outliers on the statistics, I think filtering strategies should be touched on in this manuscript. All the individual pieces are there to assert a solid filtering pipeline and demonstrate how it improves data statistics.

- p.6, line 34: Why is the bias related to the airborne campaign rather than perhaps a bad $dh/dt$ correction?

- p.6, lines 31–32: The method presented does not derive a surface slope (but rather a linear slope of elevation residuals), so where is this coming from?

- p.19: It should be clarified that the footprint diameter is the beam-limited footprint. I would like to see when the rest of the satellites are planned for launch in this table as well.

---

## Author Response (AR1)

**Sentinel-3 Delay-Doppler Altimetry over Antarctica Response to Reviewers**

**5 Reviewer 1**

This paper presents some of the first results from Sentinel 3's doppler focused radar altimeter over Antarctica. It describes the instrument and processing strategies, and it gives evidence that the data have decimeter precision over flat targets, and up to a few meters precision over sloping surfaces.

10

The paper is well written and needed very little editing. Its introduction to Sentinel-3 data should be valuable to the research community, and it makes the case well that the data are useful. The paper is somewhat weak in the analysis (rather than illustration) of the data.

- 15 I would have liked to see the elevation error magnitudes explained by mechanisms, rather than just quantified by region: it appears that even in the DML and Wilkes Land areas, the errors are multimodal, with a narrow distribution in flat areas and a wide distribution in areas with high relief. Were I using the data, I would want to assign error estimates to the data based on, for example, the local surface slope or the footprint- scale relief. Calculating errors based on histograms for areas hundreds of km on a side does not do much to help me do this. Likewise, it would be useful if the authors could explore a quantitative connection between the roughness differences at Dome C and Vostok and the biases, and perhaps include the other sites in this
- analysis. Without understanding the roughness at the DML and Wilkes sites, I don't know if the roughness difference between Vostok and Dome C is significant.

We agree with the reviewer's point and, as requested, we have added a new investigation of the impact of surface slope on our 25 error estimates. In order to do this, we have expanded our previous site-specific evaluation to cover the whole ice sheet. We have then compared the error magnitude to surface slope in order to assess the relationship between the two, and to evaluate the impact of increasing slope upon the accuracy of the Delay-Doppler altimeter measurements. We have also undertaken a

continental-scale cross-over analysis, which we use to assess the precision of the measurements as a function of surface slope. As requested, we have quantified the roughness – based upon our IceBridge profiles – at both Dome C and Vostok, and show
that this is ~4 times greater at Dome C. We have not repeated this type of analysis at the coastal sites because this analysis is specifically designed to investigate whether the relative bias at \*low slope\* sites can be explained by differences in the small-scale surface roughness. This is not relevant to the other sites because (1) there is no significant bias, and (2) they are not flat (at long wavelengths), and so at these more complex sites the radar response is a convolution of a broad range of topographic wavelengths and we believe it does not make sense to consider the small-scale topography in isolation.

My other concerns about the figures. Fonts are consistently too small, and lines are consistently too fine. The authors should print out their paper and try to read the figures!

5 We have redrawn most of the figures in the revised manuscript and have made sure that we increase the font and line size so that the figures are now much clearer. Further details are provided below in the response to Reviewer 2's comment 3, related to the figure presentation.

Figure 2: Panels d and e show the same thing as b and c. I recommend just showing just one of these profiles. The green bars 10 do not seem to show anything different from one part of the pass to another. I would recommend showing a histogram of the data contained in the bars (i.e. a histogram of the 400-m standard deviations).

As suggested, we have condensed this figure by replacing the profiles of the standard deviation - the green bar panels - with histograms of the data instead. We do, however, prefer to still keep both ground tracks because we want to show that the 15 analysis holds for more than one track, and in particular is independent of a specific viewing geometry.

Figures 3 and 4: It is hard to tell if the DEM spans the axes, or if there is extra white space around the DEM surface. The elevation axis is illegible, as are the labels on the inset panels.

20 We have revised these figures to address the points raised here, and to make them generally easier to interpret. We have replaced the 3-d DEM plots with 2-d versions overlaid upon MOA. We have also increased the font size. See also our response to the related comments by Reviewer 2 below.

Figure 3: The span of the colour scale for the coloured points is too large, and should be reduced to the span of the residuals.

As requested, we have reduced the span of the colour scale to +/-1 m.

Figure 5: I could not find a description of the differences between panels A-C, and between panels E-G, either in the text or the caption. Are there multiple profiles in d and h? The lines and points in D and H are too small, and the axis label font sizes are unreadable.

30

25

This was an omission on our part, thanks for spotting it. We have added details in the caption to explain that the profiles relate to the three different sub-tracks resolved within the ATM swath. We have also redrawn the figure with larger points, lines, and axis labels.

**Reviewer 2**

**5**

This manuscript assesses the quality of Sentinel-3 delay/Doppler radar altimetry over Antarctica through analysis of: precision at Vostok Subglacial Lake and Dome C; accuracy at Vostok, Dome C, Dronning Maud Land, and Wilkes Land; and elevation change across the continent. The authors find that Sentinel-3 altimetry achieves a precision of <0.1 m over flat topography, has a mean bias <0.2 m in the flat interior and <1.5 m in complex coastal terrain, and can reproduce known patterns in elevation-10 change rate at the continent scale. The paper represents the first use of Sentinel-3 altimetry data over an ice sheet, is generally well-written, and will be of significant value to the community as an introduction to the capabilities of this Sentinel mission, but there are three major weaknesses that should be addressed before publication:

1. Limited Analysis: While the overall analysis presented in the manuscript is sound, it is also much more limited than I would expect from a rigorous treatment of altimetry uncertainty. The authors only assess precision in two flat, interior locations, yielding a "best-case" estimate, while opting not to perform a continent-wide crossover analysis, which would provide the readers with much better insight how and why precision varies. Crossover analysis is a standard technique in satellite altimetry error analysis and should be included in any rigorous assessment.

- 20 We realise that our motivation for starting our analysis at these "best case" sites was not clear, and so we have added further text to clarify this point. The analysis at Lake Vostok was not supposed to quantify measurement precision across the entire continent. Rather it was meant as an initial analysis of the instrument precision, in order to characterise the influence on the SAR measurements of factors such as radar speckle and retracker imprecision.
- 25 This type of analysis to quantify the shot-to-shot noise we believe is best done in the absence of additional complicating factors such as complex topography, and as the reviewer points out is not indicative of measurement precision in more challenging regions. It is, however, a useful benchmark as it can be compared to similar analyses of other satellite missions that have previously been conducted at Lake Vostok.
- 30 We do agree with the reviewer's comment that a single-cycle cross-over analysis was an obvious omission from the original manuscript, and would be a useful addition. We have therefore undertaken a continent-wide cross-over analysis for two full cycles of data and used this to investigate the relationship between the magnitude of the cross-over differences and the ice sheet surface slope. As well adding new text, we have also added a new figure and a table to report these cross-over results.

In a similar vein, there are only four regions included in the accuracy analysis, when Operation IceBridge has surveyed across the continent. It is not clear why this assessment was so limited in scope, when including the rest of the Operation IceBridge airborne laser altimetry dataset would help illuminate the causes of error in the Sentinel-3 altimetry and provide the community with much more direction as to where and how to use this new dataset.

**5**

We had initially focussed on specific sites because we wanted to provide a detailed analysis over different categories of topographic terrain. We do, however, agree with the reviewer's point here, and so we have now (1) added an evaluation that utilises the continent-wide IceBridge record, and (2) analysed the relationship between the magnitude of the elevation differences and the ice sheet surface slope. We have added a new figure and supporting text to discuss these results.

10

30

Finally, the analysis of elevation-change rate is entirely qualitative; while I certainly understand that the processing chain is still in development, there should be some quantitative analysis of the result if it is to be included in the manuscript, such as a comparison to CryoSat-2 derived elevation-change rates over the same time window. At the very least, a map of "known" dh/dt from CryoSat-2 or ICESat should be shown in conjuncture with the Sentinel-3 map to show the patterns are generally similar.

15 similar

To address this point, we have expanded the quantitative description and scope of this section by (1) reporting rates of elevation change close the grounding lines of Amundsen Sea glaciers, and (2) adding an additional quantitative analysis of localised ice sheet lowering in East Antarctica, which we believe is likely to be the signature of subglacial lake drainage event. Together

- 20 these provide a quantitative, if preliminary, demonstration of the capabilities of Sentinel-3 for mapping glaciological signals of change. We would, however, prefer not to give a detailed quantitative inter-comparison between datasets, because we feel that the Sentinel-3 record is still too short for this type of analysis. Also, given the comments about efficient use of figure space, and the fact that numerous published maps of dz/dt are readily available within the literature, we believe that it is better to refer the reader to this published material, rather than to use more space reproducing them here. We have added more text
- 25 to point the reader to published baseline datasets and figures that can be used for this purpose.

2. Data and Method Detail: There are quite a few pieces of information missing with respect to the data sources used and the methods applied. There is a long list of corrections applied to the altimetry data, but no citations that provide the details of the models. There is no data citation related to the ATM laser altimetry (though there is for the Riegl data), which prevents the reader from knowing what level of data processing was used or other important details for understanding cm-scale accuracy. It appears to be L2 data; why was L2 chosen over L1b (full-swath)? Are the same corrections with the same models applied to the airborne altimetry as to the satellite altimetry? What are the dates of the Operation IceBridge flights (which is needed

differences for the satellite-airborne analysis requires some more detail: how is a 1 km DEM being used to correct for spatial 4

understanding the potential impact of the dh/dt correction)? Last, the method for correcting space and time elevation

separations of <100 m? What is the error associated with the dh/dt correction? What is the magnitude of these corrections (mm? cm? m?)? Also, at a basic level, are the results presented as hsat – hair or hair – hsat? The former is typical, but the manuscript implies the latter ("IceBridge-Sentinel-3 measurement pair"); resolving this issue is fundamental to the analysis of why the bias exists (p.7, lines 7–9).

5

10

15

20

As requested, we have added considerable extra detail to the description of our data are methods, namely:

- We have added references to describe where details of the methods and models used to generate the instrument and geophysical corrections can be found, together with a reference for the ATM laser altimetry. We choose to use the corrections supplied with, or applied within, each of the satellite and airborne products because the corrections will be instrument dependent. For example rader we believe that the teams that one rade and distribute the
- be instrument dependent for example radar vs laser and we believe that the teams that operate and distribute the data are best placed to select the most appropriate models to use.
  We have provided additional clarification that we use the Level 2 ATM data, because our understanding is that this
  - 2. We have provided additional characteristication that we use the Lever 2 ATM data, because our understanding is that this is effectively a smoothed version of the L1b, which both reduces the impact of uncorrelated shot-to-shot noise, and also brings the measurement cell closer to the resolution of the SAR footprint. We select this because we want the validation data to be as accurate, and as similar in scale to the data being validated, as possible.
  - 3. We have added details of the dates of the individual IceBridge campaigns at each site, which are now included in Table 3.
  - 4. We have provided further detail relating to the correction of spatial and temporal elevation differences, both in terms of the method used for the spatial correction (a bicubic interpolation based upon the surrounding 4x4 pixels of the 1 km DEM), and the magnitude and implications of the temporal correction (< 1cm/yr at Vostok, Dome C and Dronning Maud Land, but larger 8 cm/yr at Wilkes land).
  - 5. We have clarified that the results are indeed the typical hsat hair, and made sure this is clear and consistent within the manuscript.
- 25 **3. Figure Presentation:** The figures could be greatly improved to make them easier to understand and to use journal space better. Cyan and yellow for groundtracks on Figures 1, 3, and 5 are hard to see, printed or on a screen. I am not sure why the maps on Figures 2, 3, and 4 are plotted as 3D perspective plots; this style makes it harder to understand the spatial context, particularly how the tracks align with easting and northing (which is important since this is how the transects are plotted). The axis and legend labels on all figures are unreadably small and the line widths are too small. I am not convinced that Figures 2c
- 30 and 2e add to the discussion over a single histogram of uncertainty in 400 m along-track bins across the entire elevation range. The histograms and cumulative distribution functions in Figures 3b/3c and 3e/3f can likely be combined into a single panel. The authors should rethink how the each of the figures is presented to make the best use of space and to ensure they are fully legible at production size.
- 35 As requested we have undertaken a thorough revision of the figures. We have (1) removed the 3-d perspective plots in Figures 3 and 4, and replaced them with 2-d plots overlaid upon MOA, which hopefully makes them more easily interpretable, (2) increased the size of axis labels and line widths, (3) removed the along-track profiles in Figures 2c and 2e, and replaced them with single histograms, (4) combined the histograms and cumulative distribution functions into a single panel in all relevant figures, and (5) condensed Figures 3 and 4 into a single figure. In a few cases, we have kept the ground track lines intentionally
  - 5

thin, because we tested using a thicker line and the figures became too cluttered and difficult to interpret. Our intention is that the priority for these figures is to clearly see the elevation differences, and that the track lines are purely in the background for reference.

**5**

**Specific comments**

• p.1, line 13: "the first step towards a new era" is a bit over the top. The Antarctic coast is already blanketed in better altimetry (that includes along-track delay/Doppler processing). Please tone down these statements throughout (especially the use of 10 "novel").

Our intention here was to refer to the fact that Sentinel-3 represents a new class of \*operational\* altimetry satellites (which is important for the purpose of moving to more long-term, systematic monitoring), rather than the novelty of the delay-doppler system (which as the reviewer highlights, has already been accomplished by CryoSat-2). However, we appreciate that our

15 wording was not clear and so we have revised the text accordingly. As requested, we have also toned down similar statements throughout the manuscript.

• p.1, line 20: "accuracy decreases slightly" is an understatement, given most of the measurements are up-to an order of magnitude less accurate.

**20**

We agree this wording was too strong and have revised the text accordingly.

• p.2, line 16: the rest of the details of what controls pulse-limited footprint should go in this parenthetical statement (i.e., satellite altitude, pulse length).

**25**

We have added these details as requested.

• p.3, lines 5-7: Figure 1 should have the SARIn/LRM mode mask on it to show where Sentinel-3 altimetry will make the biggest improvement compared to existing datasets (i.e., where Sentinel-3 covers LRM areas). It also should show the latitudinal limit of Sentinel-3.

30

We have added the CryoSat-2 mode mask boundary to Figure 7 as we felt it was more useful to show the location of it here, relative to the signals of elevation change. The latitudinal limit is also clear in this plot and we have stated this within the figure caption.

• p.4, lines 12–14: The rest of the study only uses the TCOG retracker; I think adding some of the statistics from the other retrackers to Table 2 would be a fantastic addition and helpful for the community.

5 As requested, we have added statistics from the other retrackers into Table 3. We have also added a more complete description of the method in Section 3.

• p.4, lines 23-24: I would like to see the % of data points removed by this filtering step.

10 As requested, we have added statistics relating to the percentage of points removed, which is ~3% and ~3.7% for the DEM and slope correction filtering, respectively.

• p.4, line 27: This date range is insufficient given the dh/dt correction. Please provide exact flight dates, either here or in Table 2.

15

As requested, we have added the dates of the airborne campaigns for each site into Table 3.

• p.4, line 28: The along-track sampling of ATM L2 data is 0.25 s, which results in variable along-track distance spacing due to changes in aircraft velocity.

20

Thanks for the clarification, we have revised the text to make this point clear.

p.5, line 3: Why was the single closest measurement chosen, instead of some sort of average of all the points within the approximate SAR footprint? Given that surface roughness is later suggested as the cause of bias and that lidar can easily
 resolve surface roughness, it seems appropriate to try a less simplistic approach that can account for surface roughness.

During the development stage, we did test several different approaches to deriving the IceBridge validation measurement.

After testing, we selected our chosen approach because (1) we believe that it is less affected by sampling bias than the method suggested above, because we require that the IceBridge validation measurement is close to the centre of the satellite footprint,

30 whereas the method above can give poor agreement between the satellite and airborne measurements simply because the IceBridge tracks only cross one extreme of the ~ 300 x 2000 m SAR footprint, and (2) it provides a larger validation dataset – and therefore more robust statistics – than another alternative that we tested, which was based on bilinear interpolation of the IceBridge measurements that bracketed each altimeter record. We have added text to discuss this point and justify our approach.

• p.5, lines 8-10: A few words about how the distributions deviate from normal would be appreciated here. I am also somewhat worried about the use of median absolute deviation given that it is >20x smaller than the standard deviation in coastal areas, demonstrating that outliers are a significant problem for Sentinel-3 altimetry. One path forward would be to suggest a filtering method for Sentinel-3 users could apply that would bring the two statistics closer to one-another.

5

We agree that the presence of outliers has a large effect on increasing the standard deviation relative to the MAD. This is precisely why we choose to primarily report the MAD, because for non-normal distributions this statistic provides a more representative measure of the mid-point of the dispersion, which we believe is a more useful metric for the reader. Similarly,

- 10 we also report the cumulative distribution because we believe it is more helpful to know that a certain percentage of data are within a particular elevation threshold of the validation data. Nonetheless, we emphasise that we do provide both the MAD and the standard deviation within Table 2, so that the reader has full visibility of both statistics. We have also now added further text to clarify why we choose the MAD metric, and why it is appropriate for non-normal distributions. Furthermore, we have added a supplementary analysis to suggest how filtering based on the deviation from a pre-existing DEM can improve
- 15 the Sentinel-3 minus IceBridge elevation differences, although we reiterate that developing these strategies is not a priority for this work, and that the most appropriate approach will depend upon each individual application.

• p.5, lines 3-4: What % of IceBridge records were removed by this filtering step?

This step removed 7.3 % of the total Antarctic IceBridge dataset. We now report this within the manuscript. 20

• p.5, line 22: It should be noted that you are using 1σ as the definition of precision.

**We have added text to clarify this point, as requested.**

**25**

• p.6, lines 4-5: While it may be true, I do not think that the conclusion that the SAR waveform leading edge is insensitive to penetration is defensible given the analysis presented in the manuscript. Perhaps it is the Sentinel-3 sampling geometry that limits observed variability, or, even more simply, there was not much surface variability during the one-year period of observations. This statement should be removed, or alternatively there needs to be a thorough comparison between Sentinel-3

30 SAR and CryoSat-2 LRM mode using a similar retracker over the overlapping period to demonstrate that SAR processing (specifically) reduces sensitivity to penetration and subsurface scattering.

We agree with the reviewer that the conclusion about the leading edge was not supported by the analysis that we had presented here. We had confused work based on CryoSat-2 SAR data from a different study, which is not part of the current analysis, and so we have removed this sentence from the manuscript.

5 • p.6, lines 8–9: A "slope" correction, in its original meaning, is not applied in this processing chain; it is a "relocation" correction.

We agree this was not the most appropriate term, and have reworded the text to correct this point.

10 • p.6, lines 28–29: Given the importances of these outliers on the statistics, I think filtering strategies should be touched on in this manuscript. All the individual pieces are there to assert a solid filtering pipeline and demonstrate how it improves data statistics.

As suggested, we have added a supplementary analysis of how filtering based on the deviation from a pre-existing DEM can improve the Sentinel-3 minus IceBridge elevation differences, although we do reiterate that developing these strategies is not a priority for this work.

• p.6, line 34: Why is the bias related to the airborne campaign rather than perhaps a bad dh/dt correction?

20 The dh/dt correction at Dome C is < 1 cm/yr, and so we think it is unlikely that it would be in error by this magnitude. We have clarified in the text that the magnitude of the correction is only small. However, we do also acknowledge that there may be some small contribution from the dh/dt correction and so we have also added text to cover this point.

• p.6, lines 31–32: The method presented does not derive a surface slope (but rather a linear slope of elevation residuals), so 25 where is this coming from?

Yes, thanks, we agree that this was mistaken wording here and have corrected it in the revised manuscript.

• p.19: It should be clarified that the footprint diameter is the beam-limited footprint. I would like to see when the rest of the satellites are planned for launch in this table as well.

As requested, we have clarified that this refers to the beam-limited footprint diameter, and we have also added the planned launch dates for Sentinel-3C and Sentinel-3D.

|      | Sentinel-3 Delay-Doppler Altimetry over Antarctica                                                                                                                                                                                                                                                                              |                           | Formatted: Tab stops: Not at 12 cm                                                                                                                                                                                                                   |
|------|---------------------------------------------------------------------------------------------------------------------------------------------------------------------------------------------------------------------------------------------------------------------------------------------------------------------------------|---------------------------|------------------------------------------------------------------------------------------------------------------------------------------------------------------------------------------------------------------------------------------------------|

[revised manuscript text omitted]

elevation measurements with a resolution of ~ 300 m along-track by ~ 1.6 - 3 km across-track, depending upon the surface roughness (Chelton et al., 1989). To date, the focus of Sentinel-3A exploitation has been upon retrievals over ocean and inland water surfaces, with early studies demonstrating its capability to retrieve fine-scale (~20 km) oceanographic features (Heslop et al., 2017), to increase the quality of river level and discharge estimates in Central Africa (Bogning et al., 2018) and to resolve near-coastal sea surface heights (Bonnefond et al., 2018).

5

Over ice sheet surfaces. Sentinel-3 is unique among altimeters, because it operates in Delay-Doppler mode across all regions. This mode of operation contrasts with CryoSat-2, which operates in Low Resolution Mode over the interior of each ice sheet and SAR interferometric mode at coastal locations. As a consequence, although no interferometric information is available to

- 10 aid Sentinel-3 retrievals around the ice sheet margins, high resolution measurements are for the first time routinely acquired throughout the ice sheet interior. Given these different operating modes across both inland and coastal ice sheet regions, together with the future longevity of the EU Copernicus programme of operational satellites, it is imperative that early assessments of the accuracy and precision of the instrument are made over ice sheet surfaces, to establish the basis for glaciological applications of these data. Here we provide a first evaluation of Sentinel-3 Delay-Doppler measurements over
- Antarctica, to determine its utility for monitoring ice sheet surfaces. 15

**2. Study Sites**

To evaluate the performance of the Sentinel-3 Delay-Doppler altimeter across a range of topographic regimes, we selected four study sites across Antarctica for detailed analysis (Figure 1). Two sites - the areas surrounding Lake Vostok and Dome C - are located within the interior of the East Antarctic Ice Sheet and are characterized by relatively simple topography. These

- sites allowed us to evaluate the performance of Delay-Doppler altimetry in regions representative of a large part of the Antarctic 20 interior. Both sites have low and relatively uniform topographic slopes, with an average and standard deviation of 0.09° and 0.05° (Lake Vostok), and 0.04° and 0.03° (Dome C), respectively, based upon a 1km Digital Elevation Model (DEM) (Slater et al., 2018. Furthermore, the flat ice surface directly above Lake Vostok represents an established validation site for new altimetry missions (Richter et al., 2014; Schröder et al., 2017; Shuman et al., 2006). To assess performance in regions of
- 25 steeper and more complex topography, we then selected two coastal sites covering parts of Dronning Maud Land and Wilkes Land (Figure 1). These locations were chosen because of the availability of airborne campaigns that could be used as independent validation. Both sites have an order of magnitude steeper and less uniform topography than the inland sites, with the mean and standard deviation of the surface slope being 0.50° and 0.94° (Dronning Maud Land) and 0.40° and 0.51° (Wilkes Land), respectively. Finally, beyond these focused, site-specific studies, we also conducted several continent-wide analyses,

30 in order to better understand the performance of the Sentinel-3A altimeter across a broader range of topographic regimes.

**Formatted: Font: Italic Deleted: the Deleted: novel Formatted: Font: Italic Formatted: Font: Italic Formatted: Font: Italic**

| Deleted: se                           |  |
|---------------------------------------|--|
| Deleted: novelty                      |  |
| Deleted: ces in                       |  |
| Deleted: on                           |  |
| Deleted: of the Sentinel-3 instrument |  |

**Deleted:**

**Deleted: performance**

| (  | Deleted: Thomas         |
|----|-------------------------|
| ~( | Formatted: Font: Italic |
| ~( | Deleted: 7              |
| )  | Formatted: Font: Italic |
| X  | Deleted: novel          |
| X  | Formatted: Font: Italic |
| Y  | Deleted: also           |

**3. Sentinel-3 Data & Processing Methods**

To evaluate the accuracy of Sentinel-3A elevation measurements, we begun by analysing 14 cycles of Sentinel-3A SRAL data acquired between December 2016 and December 2017. Our processing followed a standard chain, beginning with the 20 Hz waveform data provided by the European Space Agency (ESA) within their freely-distributed 'enhanced' data file, and

- 5 generated using their Processing Baseline 2.27. Firstly, an estimate of the waveform noise was made from the mean power of the lowest six waveform samples, and waveforms where this value exceeded 0.3 of the maximum recorded power were rejected and not passed to the subsequent processing. Next, each remaining waveform was oversampled by a factor of 100 using a spline interpolation and then the leading edge of each waveform was identified based upon the first set of waveform samples that satisfied the following criteria, (1) a normalised power that exceeded the noise floor (defined as being 0.05 above the mean
- 10 normalised power of the lowest 6 samples), (2) a change in normalised power from the noise floor to the next waveform peak that was greater than 0.2, and (3) that ended with a defined waveform peak (such that there was a decrease in power at delay times beyond the peak location). Each waveform that had a leading edge satisfying these criteria was then retracked using several empirical retrackers, namely a Threshold on the offset Centre Of Gravity amplitude (TCOG) (Wingham et al., 1986), a Threshold First Maximum Retracker (TFMRA) (Helm et al., 2014), and a maximum gradient of the first leading edge
- 15 retracker (*Gray et al., 2015*). For the first two solutions, a threshold of 50 % of the leading edge power (defined as the leading edge maximum power minus the noise floor) was used, with the aim of providing a stable retracking point across both low slope and more complex topographic surfaces. More specifically, we chose this mid-power threshold as a balance between minimising the sensitivity to noise at the start of the waveform leading edge, and also to reducing the impact of radar speckle (which is more apparent near to the waveform peak due to its multiplicative nature). For the majority of this study, we have
- 20 focused on reporting results produced by the TCOG retracking, because of the continuity it provides with the ground segments of past European Space Agency (ESA) missions, and the broadly consistent results between all three of the retrackers tested. However, for completeness we do also report statistics from all retrackers within our independent validation exercise.

After retracking, Level-2 instrument and geophysical corrections were applied to each range measurement to account for the

- 25 distance between the antenna and satellite centre of mass, dry and wet troposphere delays, ionosphere delays, solid Earth tide, ocean loading tide and Polar tide, plus ocean tide and the inverse barometer effect over floating ice. These corrections are all included within the enhanced data product, and further details can be found within the product specification (Sentinel-3 Core PDGS Instrument Processing Facility (IPF) Implementation Product Data Format Specification, 2015). These geophysical corrections are provided at 1 Hz sampling, and so we used linear interpolation to resample these fields to the native 20 Hz rate
- 30 of the altimeter measurements. The echoing point was then relocated to the point of closest approach (*Roemer et al., 2007*) within the SAR beam footprint using a DEM derived from 7 years of CryoSat-2 data (*Slater et al., 2018*), with echoing points that were relocated by more than ~ 8 km, and therefore at the edge of antenna beamwidth removed (~ 3.7 % of data, based on

**Deleted: Altimeter**

Deleted: First, unusable waveforms, where no clear leading edge could be identified, were rejected and not passed to the subsequent processing. Then, each remaining waveform was retracked using a variety of empirical retrackers including a Threshold on the offset Centre of Gravity amplitude (TCOG) [Wingham et al., 1986], a Threshold First Maximum Retracker (TFMRA) [Helm et al., 2014], and a maximum gradient of the first leading edge retracker [Gray et al., 2015]. For the first two solutions, a threshold of 50 % of the waveform power was selected with the aim of providing a stable retracking point across both low slope and more complex topographic surfaces. More specifically, we chose this mid-power threshold as a balance between minimising the sensitivity to noise at the start of the waveform leading edge, and also to reducing the impact radar speckle, which is more apparent near the waveform peak due to its multiplicative nature. For the majority of this study, we focused on results produced by the TCOG retracking because of the continuity it provides with the ground segments of past European Space Agency (ESA) missions, and the broadly consistent results between all three of the retrackers tested

Deleted: each waveform was normalised and oversampled by a factor of 100 using a spline-based interpolation. A...n empirical estimate of the waveform noise was then ... ade from the mean power of the lowest six6...waveform samples, and waveforms where this value exceeded 0.3 of the maximum recorded power were rejected and not passed to the subsequent processing. Next, each remaining waveform was oversampled by a factor of 100 using a spline interpolation and then the leading edge of each waveform was identified based upon the first set of waveform samples that satisfied the following criteria, (1) a normalised power greater than ... hat exceeded 0.05 above the noise floor (estimated effined as being 0.05 aboveas...the mean normalised power of the lowest 6 samples). (2) a change in normalised power from the noise floor to the leading edge...ext waveform peak that was peak...greater than 0.2, and (3) that ended with a defined waveform peak (such that there wai ... a decrease in power at delay times beyond the peak location). Each[10]

F111

**Formatted**

| $\sim$ |                                                                                                                                    |
|--------|------------------------------------------------------------------------------------------------------------------------------------|
|        | Deleted: includeeport statistics from all retrackers within our independent validation using airborne altimetryxercise (Table 112] |
| )
( | Deleted: ¶                                                                                                                         |
|        | Deleted: regarding theiran be found within the product specification (Sentinel-3 Core PDGS Instrument Processing Facility)         |
| -(1    | Formatted: Font: Italic                                                                                                            |
| -(1    | Deleted: although t                                                                                                                |
| .(     | Formatted: Font: Italic                                                                                                            |
| (      | Deleted: Thomas                                                                                                                    |
| (      | Formatted: Font: Italic                                                                                                            |
| (      | Deleted: 7), with . E [14]                                                                                                         |
| -(     | Deleted: Relocated points falling [15]                                                                                             |
| ľ      | Deleted: placing themt the edge of antenna 3dB beamwidth,oroutside of the ~ 0.3 by 18 km SAR beam footpript61               |

[revised manuscript text omitted]

have been estimated to have a vertical accuracy and precision of 7 cm and 3 cm, respectively (*Martin et al., 2012*). At our two coastal sites, where ATM measurements have not been acquired, we instead used RLA acquisitions. This instrument has a smaller ground footprint of 25 m along track by 1 meter across track, and a slightly larger reported accuracy of 12 cm (*Blankenship et al., 2013*).

5

To compute elevation differences between our Sentinel-3A and IceBridge datasets, we identified IceBridge records within a 200-meter search radius of each satellite measurement. Where multiple IceBridge records existed within the search radius, we selected the closest measurement. Alternative methods for selecting IceBridge measurements were also tested\_such as bilinear interpolation of multiple surrounding measurements, but this approach produced a less comprehensive set of comparison points.

- 10 from which to generate our assessment statistics. As part of this process, we identified and removed anomalous IceBridge elevation records that deviated by more than 100 meters from an independent DEM (*Slater et al., 2018*). This step removed 7.3% of the total Antarctic IceBridge dataset. We then corrected for elevation differences arising from the spatial and temporal separation of the satellite and airborne measurements. In the case of the former, we constructed a bicubic interpolation of the surrounding 4x4 pixel area of the DEM surface, and used this to estimate the difference in elevation between the satellite and
- 15 airborne measurement locations, For the latter, we used an estimate of the local rate of elevation change (McMillan et al., 2014), The magnitude of the elevation change correction is small (< 1 cm yr1) at the Vostok, Dome C and Dronning Maud Land sites. At our Wilkes Land site, the magnitude of the correction is larger (8 cm yr1) and it is therefore possible that inaccuracies in the correction could contribute, in part, to the differences between the airborne and satellite measurements at this site. For example, a 10 % error in the correction over a 5-year period would equate to a 4 cm error in the corrected
- 20 IceBridge elevation. Finally, we computed the Sentinel-3 minus IceBridge elevation difference for each measurement pair, and so generated a set of statistics for each study site (Table 3). Because the differences, particularly at coastal sites, are not normally distributed, and exhibit higher clustering around the central value, together with a greater proportion of outliers, we principally use the median and median absolute deviation (MAD) from the median as measures of the bias and dispersion, respectively. We choose to use the MAD, because for non-normal distributions this statistic provides a more representative
- 25 measure of the mid-point of the dispersion. We do, nonetheless, report both the MAD and the standard deviation, within Table 3

**5.2 Evaluation at Inland Sites**

At the inland sites of Lake Vostok and Dome C, we find very good agreement between the Sentinel-3A and airborne datasets / (Figure 4 and Table 3). At Lake Vostok, the median bias between our TCOG solution and the airborne data is 1, cm, and the solution.

30 MAD dispersion of the differences is 13 cm. At Dome C, the bias is larger (20 cm), but the dispersion of the differences is smaller (6 cm). The differing bias between the two sites is investigated in more detail in Section 5.4. Comparing the results from the different retrackers, we find variations of approximately 10-30 cm in the median bias, which reflects differences in

**16**

**Deleted: offer**

| -(  | Formatted: Font: Italic                                                                                                                                                                                                                            |
|-----|----------------------------------------------------------------------------------------------------------------------------------------------------------------------------------------------------------------------------------------------------|
| 4   | Formatted: Font: Italic                                                                                                                                                                                                                            |
| Å   | Deleted: validation                                                                                                                                                                                                                                |
| 1   | Deleted: 1                                                                                                                                                                                                                                         |
| 1   | Deleted: Other                                                                                                                                                                                                                                     |
| Ì   | Deleted: (for example                                                                                                                                                                                                                              |
| Â   | Deleted: )                                                                                                                                                                                                                                         |
| 4   | Deleted: although                                                                                                                                                                                                                                  |
| (   | Deleted: these typically                                                                                                                                                                                                                           |
| -(  | Deleted: and therefore a smaller dataset                                                                                                                                                                                                           |
| -(  | Deleted: Next                                                                                                                                                                                                                                      |
| -(  | Deleted: Thomas                                                                                                                                                                                                                                    |
| Y   | Formatted: Font: Italic                                                                                                                                                                                                                            |
| Y   | Deleted: 7                                                                                                                                                                                                                                         |
| -(  | Deleted: ,                                                                                                                                                                                                                                         |
| -(  | Deleted: using the difference in DEM elevations at the measurement locations and                                                                                                                                                            |
| Y   | Deleted: Malcolm                                                                                                                                                                                                                                   |
| Y   | Formatted: Font: Italic                                                                                                                                                                                                                            |
| Y   | Deleted: , respectively                                                                                                                                                                                                                            |
| Ì   | Formatted: Superscript                                                                                                                                                                                                                             |
| (   | Deleted: between                                                                                                                                                                                                                                   |
| - ( | Deleted: (Table 2)                                                                                                                                                                                                                                 |
| -(  | Deleted: were                                                                                                                                                                                                                                      |
| Å   | Deleted: Additional statistical measures are, however, included                                                                                                                                                                                    |
| Å   | Deleted: Table 2                                                                                                                                                                                                                                   |
| λ   | Deleted: 1                                                                                                                                                                                                                                         |
|     | Deleted:
To establish an independent evaluation of the accuracy of the
Sentinel-3 altimeter over ice sheets, we compared data from the first
year of routine operations to Operation IceBridge airborne elevation
measurements. |
| Å   | Field Code Changed                                                                                                                                                                                                                                 |
| Å   | Deleted: Figure 3)                                                                                                                                                                                                                                 |

[revised manuscript text omitted]

**Deleted however**

**Formatted: Font: Italic**

Deleted: We then compared the magnitude of the median absolute elevation differences to the magnitude of the surface slope (Figure 5). For this analysis, surface slope was derived from an independent CryoSat-2 DEM (T. Slater et al. 2018) At the continent scale we find a Sentinel-3 minus IceBridge median elevation difference of 0.06 m, and MAD of the elevation differences of 1.06 m. It is important to note, however, that these do not represent an unbiased sample of the total ice sheet distribution, because the IceBridge surveys are more frequently acquired across steep and more complex ice margin topography (Figure 5).

The median slope at IceBridge validation points is 37% higher than the median slope of the ice sheet.

| Formatted                                                                                                                                                          | ]               | [54]             |
|--------------------------------------------------------------------------------------------------------------------------------------------------------------------|-----------------|------------------|
| Deleted: less than46                                                                                                                                               |                 | [55]             |
| Deleted:degrees                                                                                                                                                    |                 | [56]             |
| Deleted: approximately 824% of the ice sheet area, the n absolute difference between Sentinel-3 and IceBridge is less the order metre, whereas f, whereas f | nedia
than   | an
of
[57] |
| Deleted: degree                                                                                                                                                    |                 |                  |
| Deleted: , which is mostly found across the Antarctic Penir transantarctic mountains,the difference increases to ~the or                                    | ısula
der    | and
[958]     |
| Deleted: It should also be e noted that these statistics re                                                                                                        | flect           | [59]             |
| Deleted: currently being tested and are expected                                                                                                                   |                 |                  |
| Deleted: 1                                                                                                                                                         |                 |                  |
| Deleted: i perhaps moreess unxpected that, for any retracker, we find a difference of ~20 cm in the bias recorded                                           | givei
d at t | n
he          |

[revised manuscript text omitted]

- *al.*, 2014), and provides an early indication that the Sentinel-3 instrument and orbital configuration is suitable for mapping changes across the low relief ice sheet interior. Across the margins of the ice sheet, even over this short time period, there is evidence that Delay-Doppler altimetry is able to map the higher, dynamically-driven, rates of elevation change that are known to be occurring in these topographically more complex regions (*Flament & Rémy, 2012; Helm et al., 2014; McMillan et al.,*
- 5 2014). Widespread elevation change is evident across the fast-flowing ice streams draining into the Amundsen Sea Sector of West Antarctica, with rates of surface lowering of 2-3 m yr1 close to the grounding line of Pine Island Glacier and 4-5 m yr1 upstream of the grounding lines of the Thwaites and Smith Glaciers. In comparison to inland regions, the coverage is generally poorer across the steeper ice margin regions and the more mountainous terrain of the Antarctic Peninsula. This is expected given the lower precision of elevation measurements in these locations, the wider track spacing, and the short time period over
- 10 which the trends are being computed. Additionally, it is important to note that the current approach to performing the waveform windowing and SAR multi-looking within the Level-1B processing chain of the ground segment is not fully optimized for these challenging ice regions. Further refinements to this processing step, namely to adjust the windowing during the Doppler beam stacking to account for large variations in the satellite tracker range, are currently being implemented, and are expected to deliver future improvements in data retrieval in these regions. Based upon this preliminary assessment, however, there is
- 15 good reason to expect that Sentinel-3 Delay-Doppler altimetry will prove to be an effective tool for mapping ice sheet elevation change.

Finally, we investigated the capability of SAR altimetry to make precise measurements of surface elevation change within the *j* inland regions of the ice sheet, where previously only Low Resolution Mode observations have been available. Specifically,

- 20 we focused on a small region (of the order of tens of kilometres) of anomalously high elevation changes at a location within the interior of the East Antarctic Ice Sheet (location shown in Figure 7a). We analysed 28 cycles of Sentinel-3A data passing over this region, grouping data from all cycles within 340 m intervals along-track, and again employing a model fit approach to isolate the temporal evolution of the ice surface with a 27-day repeat frequency (*Moholdt et al., 2010; Smith et al., 2009*), On either side of this region of high elevation change, we find a high level of repeatability of the SAR elevation measurements.
- 25 giving us confidence in the precision of the instrument and our method for isolating along-track elevation changes through / time (Figure 7b). Over the feature itself, we resolve a progressive, spatially coherent movement of the ice sheet surface, with / a total lowering of ~ 1.7 metres over a period of 16 months. Transient changes in elevation at this spatial scale are widely a understood to be caused by subglacial lake drainage (Smith et al., 2009), and as such, our observations provide a first indication of the capability of Sentinel-3 SAR altimetry to systematically monitor such events.

**30 7. Conclusions**

We have undertaken a first assessment of the utility of Sentinel-3 Delay-Doppler (SAR mode) altimetry for measuring ice sheet elevation and elevation change using the standard ESA Level-1b product and our own Level-2 processing chain. Analysis

| λ    | Formatted: Font: Italic                                                                                                                                                                                                              |
|------|--------------------------------------------------------------------------------------------------------------------------------------------------------------------------------------------------------------------------------------|
| -(   | Deleted: Malcolm                                                                                                                                                                                                                     |
| -(   | Deleted: surface lowering                                                                                                                                                                                                            |
| 1    | Deleted: elevation change                                                                                                                                                                                                            |
| (    | Deleted: -                                                                                                                                                                                                                           |
| -(   | Deleted: -                                                                                                                                                                                                                           |
| -(   | Deleted: near to the termini                                                                                                                                                                                                         |
| (    | Deleted:                                                                                                                                                                                                                             |
| K    | Deleted: the high resolution                                                                                                                                                                                                         |
| (    | Deleted: mode                                                                                                                                                                                                                        |
| (    | Deleted: investigated in more detail                                                                                                                                                                                                 |
|      | Deleted: that appeared to show locally higher rates of change than its surroundings                                                                                                                                           |
| (    | Deleted: XX                                                                                                                                                                                                                          |
| /(   | Formatted: Font: Italic                                                                                                                                                                                                              |
|      | Deleted: and used an along-track method to estimate – and remove
– elevation changes due to the across-track slope, which manifest in
the elevation profiles due to the satellite drift within the orbital dead
band |
| ļ    | Deleted: The results ()                                                                                                                                                                                                              |
| //   | Deleted: show                                                                                                                                                                                                                        |
| 1    | Deleted: n either side of this feature                                                                                                                                                                                               |
| (    | Deleted: in this region                                                                                                                                                                                                              |
| ļ    | Deleted: see                                                                                                                                                                                                                         |
| ļ    | Deleted: pattern                                                                                                                                                                                                                     |
| ļ    | Deleted: ice deflation                                                                                                                                                                                                               |
| Ķ    | Deleted: the surface dropping                                                                                                                                                                                                        |
| Å    | Deleted: by                                                                                                                                                                                                                          |
| Å    | Deleted: XX                                                                                                                                                                                                                          |
| 1    | Deleted: XX                                                                                                                                                                                                                          |
| 1    | Deleted: This scale of signal is similar to multiple other cases which have been interpreted as                                                                                                                               |
| ···( | Deleted: events                                                                                                                                                                                                                      |
| X    | Formatted: Font: Italic                                                                                                                                                                                                              |
| Y,   | Deleted: .                                                                                                                                                                                                                           |
| Ì    | Deleted: We therefore suggest that this is a similar such event, and a clear demonstration of                                                                                                                                 |
| Y)   | Deleted: altimetry                                                                                                                                                                                                                   |
| Y    | Deleted:                                                                                                                                                                                                                             |

of repeated acquisitions over the Lake Vostok validation site indicates that, over the first year of routine operations, the instrument has operated with sub-decimeter precision. Through validation with airborne campaigns, we find small median biases in elevation, typically of the order 1-10 cm, at both inland and coastal sites. The dispersion of elevation residuals, measured with respect to the validation data, is of the order of 10 cm at inland sites, increasing to  $\tau_{\rm s}$  m at coastal sites with

- 5 more complex topography. This reflects the main challenges associated with processing radar altimetry data in complex ice margin regions, namely (1) reliable retracking of multipeak waveforms that arise when multiple distinct surface reflections are captured within the receive window, and (2) accurately establishing the location of the echoing point within the SAR beam footprint. These represent principle avenues of future research within the field of ice sheet Delay-Doppler altimetry. Nonetheless, the accuracy achieved in even these complex ice margin regions is encouraging, and expected to improve further
- 10 as refinements are made to the operational ground segment processing. Finally, we have shown the capability of Sentinel-3, albeit with the relatively short record of data currently available, to resolve the known signals of elevation change that currently dominate Antarctica's contribution to sea level rise, and to monitor subglacial lake activity. Together, our analysis demonstrates the early promise of Sentinel-3 SAR altimetry as a platform for long-term, operational monitoring of Earth's ice sheets.

[revised manuscript text omitted]

---

## Author Response (AR2)

**Sentinel-3 Delay-Doppler Altimetry over Antarctica**

Malcolm McMillan1, Alan Muir2, Andrew Shepherd3, Roger Escolà4, Mònica Roca4, Jérémie Aublanc5, Pierre Thibaut5, Marco Restano6, Américo Ambrozio7, Jérôme Benveniste8

5

1Centre for Environmental Data Science & Centre for Polar Observation & Modelling, Lancaster University, Lancaster, LA1 4YW, UK

[revised manuscript text omitted]